# MethNet: a robust approach to identify regulatory hubs and their distal targets from cancer data

Theodore Sakellaropoulos[1,2,5], Catherine Do[1,2,5], Guimei Jiang[1,2,5], Giulia Cova [1,2], Peter Meyn[3], Dacia Dimartino[3], Sitharam Ramaswami[3], Adriana Heguy [3], Aristotelis Tsirigos [1,2,4] ✉ & Jane A. Skok[1,2] ✉

Aberrations in the capacity of DNA/chromatin modifiers and transcription factors to bind non-coding regions can lead to changes in gene regulation and impact disease phenotypes. However, identifying distal regulatory elements and connecting them with their target genes remains challenging. Here, we present MethNet, a pipeline that integrates large-scale DNA methylation and gene expression data across multiple cancers, to uncover cis regulatory elements (CREs) in a 1 Mb region around every promoter in the genome. MethNet identifies clusters of highly ranked CREs, referred to as 'hubs', which contribute to the regulation of multiple genes and significantly affect patient survival. Promoter-capture Hi-C confirmed that highly ranked associations involve physical interactions between CREs and their gene targets, and CRISPR interference based single-cell RNA Perturb-seq validated the functional impact of CREs. Thus, MethNet-identified CREs represent a valuable resource for unraveling complex mechanisms underlying gene expression, and for prioritizing the verification of predicted non-coding disease hotspots.

Both coding and noncoding elements can drive cancer and its resistance to therapy, but coding regions, which make up a mere 2% of our genome, are typically the focus of analysis. This is because of the high cost of whole genome sequencing and the fact that coding sequences are easily identifiable and can be directly linked to changes in gene expression[1,2]. In contrast, it is difficult to connect cis-regulatory elements (CREs) in non-coding regions to their target genes as these can be located many hundreds of kilobases away on the linear chromosome. Nonetheless, noncoding regulatory elements like promoters, enhancers, silencers and structural elements cannot be ignored since noncoding variants have been shown to be more likely to contribute to disease susceptibility than non-synonymous coding variants. Furthermore, noncoding regulatory elements occupy a greater proportion of the genome compared to coding sequences, and they alter the binding

capability of factors that are the key drivers of gene regulation. In addition, and equally important, epigenetic changes that alter the ability of a transcription factor (TF) to bind a regulatory element can have the same effect[3–5].

CREs are marked by active histone marks and DNA hypomethylation and are enriched for the binding of transcription factors. Distally located CREs rely on cohesin-mediated loop formation to bring them into physical contact with the promoters of genes they control[6]. The chromatin contacts can be stable (found in all cell types), or cell-type specific with contacts mediated by cell-type specific TFs, in a CTCF dependent or independent manner. Although gene regulation commonly occurs between a single CRE-promoter pair, transcriptional control can be complicated by CRE redundancy. Indeed, we and others have shown that enhancers can control the expression of more than

[1]Department of Pathology, NYU Grossman School of Medicine, New York, NY, USA. [2]Perlmutter Cancer Center, NYU Langone Health, New York, NY, USA. [3]Genome Technology Center, NYU Grossman School of Medicine, New York, NY, USA. [4]Applied Bioinformatics Laboratories, Office of Science & Research, NYU Grossman School of Medicine, New York, NY, USA. [5]These authors contributed equally: Theodore Sakellaropoulos, Catherine Do, Guimei Jiang. ✉e-mail: aristotelis.tsirigos@nyulangone.org; jane.skok@med.nyu.edu

one gene[7,8], and similarly, promoters can act as enhancers that regulate other distally located target genes[9]. Thus, gene regulation can occur in 'hubs', which encompass multiple CREs and the promoters of their gene targets. CRE hubs connect regulatory elements that can be widely separated on the linear chromosome, but with interactions largely restricted to the same topologically associated domain (TAD), suggesting they rely on a loop extrusion mechanism for their formation[10]. Hubs are strongly enriched for promoters of target genes and super-enhancers critical for cell identity and are associated with high transcriptional activity indicating a probable important role in gene regulatory networks that control cell fate[11–14].

Understanding the mechanisms by which genes are controlled is challenging for a number of reasons. First, as mentioned above, regulatory elements can be located hundreds of kilobases away on the linear chromosome and they do not necessarily control the nearest neighboring gene. Second, even though regulatory elements and their target genes are generally in close contact in 3D space, as a result of chromatin looping, not all elements that are in contact have a functional impact on gene regulation[8]. Third, although enrichment of active histone marks, DNA hypomethylation and transcription factors are all hallmarks of regulatory elements, their presence does not imply functional impact. Finally, there is no 'one rule for all', and every gene can be controlled by a combination of regulatory elements (enhancers / silencers) and unique chromatin folding constraints (mediated by binding of CTCF and cohesin). Thus, in order to better understand mechanisms underlying transcriptional control, we need to take advantage of datasets that couple gene expression with epigenetic marks to construct functional models.

DNA methylation is an epigenetic mark associated with the regulation of gene expression[15]. It has long been appreciated that methylated CpG islands (CGI) in the promoter of a gene have a strong silencing effect, but beyond that, the functional impact of methylation on gene expression is context-dependent. Intragenic methylation occurs during transcript elongation in gene bodies to prevent the initiation of spurious transcripts[16–18], while methylation of intronic and intergenic regions impacts the activity of regulatory elements[19]. Bisulfite converted sequence-based techniques such as whole genome bisulfite sequencing (WGBS), reduced-representation bisulfite sequencing (RRBS) and array-based techniques like Illumina's Bead-Chip, have facilitated the genome-wide screening of chromatin for methylation marks, allowing for a systematic investigation of their methylation status. The ENCODE[20] and TCGA projects are particularly useful resources for identifying novel putative regulatory elements as they provide paired DNA methylation and gene expression data across multiple cell lines and patient-derived samples.

Several approaches have been proposed to investigate the role of methylation in gene regulation in a systematic way. These methods can be divided into two broad categories based on the manner in which they interrogate possible connections: (i) association mining and (ii) regression modeling. In association mining, all possible pairwise connections between candidate regulatory sites and genes are tested independently, while in regression modeling all possible associations are considered simultaneously. The association mining category includes methods like ELMER[21,22] and TENET[23], in which correlations between differentially methylated sites and differentially expressed genes in tumor versus normal samples, are tested. The regression modeling category includes methods like ME-Class[24] and the use of Random Forests (RFs) to predict whether a gene will be differentially expressed based on changes in methylation status and other chromatin features[25]. However, a limitation of this method is that the analysis is restricted to promoters and gene bodies. Other regression methods like REPTILE[26], use RFs to directly identify putative enhancers instead of modeling enhancer-gene associations. Aside from their strengths and weaknesses, all the above methods focus on a narrow region around the promoter and the gene body.

In this paper we present MethNet, a pipeline to uncover regulatory networks linking CpG sites to gene expression. In contrast to other methods, we identify regulatory elements within a 1 Mb region around the promoter of every protein-coding gene in the genome, taking into account the overall decay of regulatory connections related to the distance separation from a gene. The resulting regulatory network recapitulated known regulatory mechanisms like the silencing effect of methylation at gene promoters and the enrichment of CpG islands in CREs. Importantly, MethNet also identified potential CREs whose methylation was correlated with transcriptional activation or repression of their associated target genes. We characterized these networks and used them to identify cis-regulatory hubs involving multiple associations with a robust regulatory potential that are predictive of patient survival across all cancers as well as in particular tumor types. Promoter-capture Hi-C confirmed that highly ranked associations are mediated through physical interactions between CREs and their target gene promoters. Further, CRISPR inteference (CRIS-PRi) based single-cell RNA (scRNA) Perturb-seq confirmed that Meth-Net was able to accurately identify functional regulatory elements. Thus, MethNet is a powerful and cost-effective tool that can be used to understand the underlying mechanisms of distal gene regulation and to prioritize the verification of predicted non-coding disease hotspots.

## Results

### MethNet constructs regulatory networks using TCGA data

Most of the published epigenome-wide association studies (EWAS) have focused on links between differentially methylated regions and the expression of the closest gene. However, it is important to account for long-range interactions when modeling gene expression since regions that are distal on the chromatin fiber can be brought into close proximity by chromatin folding. Indeed, gene regulation occurs largely within highly self-interacting, megabase sized 'topologically associated domains' (TADs)[27–29], that are separated by 'insulating boundaries' enriched for CTCF and cohesin. The boundaries are functionally important as they limit inter-TAD interactions, so that enhancers predominantly contact promoters within the same TAD. TADs form via 'loop-extrusion', with cohesin rings extruding DNA until they encounter two convergently oriented CTCF binding sites, which form the base of a loop[30].

To account for this, MethNet considers as potential regulatory elements all the CpG probes that are within a 1Mbp radius of the TSS. It uses gene expression and methylation data from TCGA samples to construct predictive models that quantify the contribution of each site to a gene's regulation (Fig. 1). TCGA is the largest resource with both genome-wide gene expression and DNA methylation data from the same patient samples and is therefore well-suited for this type of analysis. Gene expression is modeled separately for each cancer-type to account for context-specificity. Elastic-net regularization is used to restrict spurious associations. A consensus network was constructed by aggregating associations across all cancers, so that robust associations found in multiple cancers were ranked higher relative to cancer specific associations and thus are less likely to correspond to false-positive results. Finally, we computed the regulatory potential of every CRE as a function of all the associations it is involved in, and characterized the epigenetic features that contribute (Fig. 1).

### MethNet identifies putative activating and repressing distal associations

The challenge for any CRE-discovery method is that the number of potential associations grows exponentially with the radius of their regulatory window (Fig. 2a). A-priori there are approximately 8 million potential regulatory associations between CpG sites in a window size of 1 Mb on either side of every protein coding promoter. MethNet uses an average of 400,000 (5%) associations per individual cancer to model the expression of every gene (more statistics are shown in

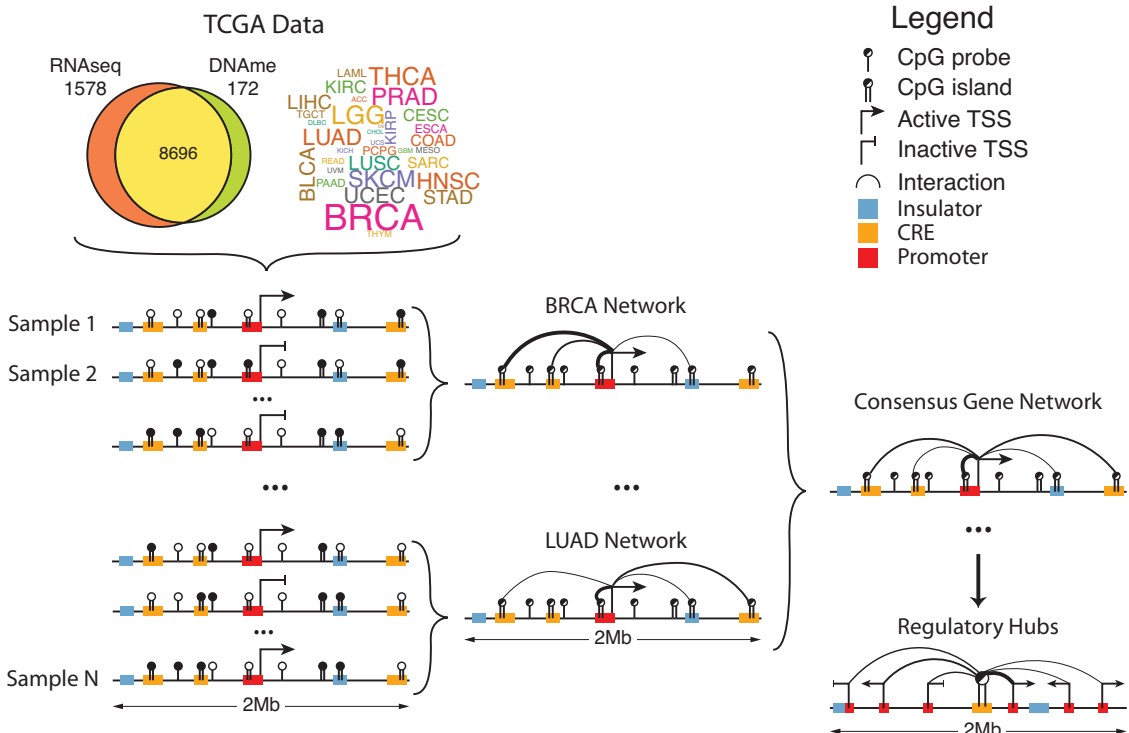

**Fig. 1 | Outline of the MethNet pipeline.** Paired RNA-seq and DNA-methylation data were accessed from TCGA. The expression of every protein coding gene across every cancer dataset was modeled as a function of the methylation status in a 1 Mb radius surrounding the gene to generate a set of regulatory networks across all cancers. These were aggregated to produce a network of robust associations that were used to identify regulatory elements and hubs.

Figure S1). A pan-cancer analysis indicated that there is strong context-specificity in the regulatory network, with up to one third (2.6 million) of all potential CREs having a putative functional impact in at least one cancer. The strength of these regulatory associations varied across different cancer types, with the majority of associations being specific to a particular cancer, consistent with lineage specific transcription factor-mediated gene regulation (Fig. 2b). The context-specificity of the regulatory networks identified are associated with varying degrees of accuracy and reliability across different cancer types. Notably, larger sample sizes enhance MethNet's capacity to uncover robust and reliable associations (Fig. 2c). This observation underscores the potential scalability of MethNet and its ability to leverage larger datasets to further elucidate the complex regulatory mechanisms governing gene expression in cancer. The results of the MethNet pipeline are shared as supplementary data on figshare (see Data Availability).

MethNet successfully recovers known regulatory mechanisms such as the well documented effect of gene silencing when a promoter is methylated (Fig. 2d). Specifically, methylation of the first CpG island upstream of the TSS has a strong negative correlation with expression. Furthermore, MethNet identifies CpG islands as being more likely to associate with and have a stronger effect on gene expression than inter-island regions in an unbiased way[31] (Fig. 2d). On average, the closer a CRE is to its target gene, the greater the probability of having an impact on transcriptional output. Importantly, however, the probability of CRE-target gene association does not diminish to zero, indicating that CRE methylation beyond the immediate vicinity of the promoter can have an impact on gene expression in a context-specific manner.

We define MethNet associations that have a negative or positive coefficient as activating or repressing, respectively. In activating associations, a gain of methylation is predicted to decrease gene expression, while in repressive associations, a gain of methylation is predicted to increase gene expression. For example, MethNet identified a CTCF binding site located 250 kb upstream of the *IFNγ* promoter whose demethylation is linked to transcriptional repression (Fig. 2e and Supplementary Fig. S2). This suggests that CTCF binding to the unmethylated DNA sequence could be acting as an insulator preventing the *IFNγ* promoter from coming into contact with elements that activate its expression. Silencing of this inflammatory cytokine could have profound effects on immune responses to cancer, and its regulation is thus important in the context of immunotherapy. In contrast, MethNet identified the promoter of a non-coding gene located downstream of *GSTT1* that when demethylated activates Glutathione S-transferase1 expression. This suggests that the promoter of the non-coding gene may be acting as an enhancer for *GSTT1* (Fig. 2f and Supplementary Fig. S3). Glutathione S-transferases (GSTs) are phase II metabolizing enzymes that play a key role in protecting against cancer by detoxifying numerous potentially cytotoxic/genotoxic compounds. These two examples highlight the complex interplay between DNA methylation and gene expression, providing valuable insight into the mechanisms underlying the regulation of cancer relevant genes.

## The regulatory potential of CREs is correlated with chromatin context and contact frequency

The number of potential CREs controlling transcriptional output varies by gene, and the number of genes controlled by a single CRE varies by element. To study individual CREs and rank them by their impact on transcriptional output we defined a metric to capture the intrinsic regulatory potential. In particular, we aggregated a CRE's contribution to the regulation of all genes in its vicinity. We quantified the importance of an association as the excess of its relative effect size with respect to a null model, where all elements have the same intrinsic potential and the only factor differentiating them is their distance to

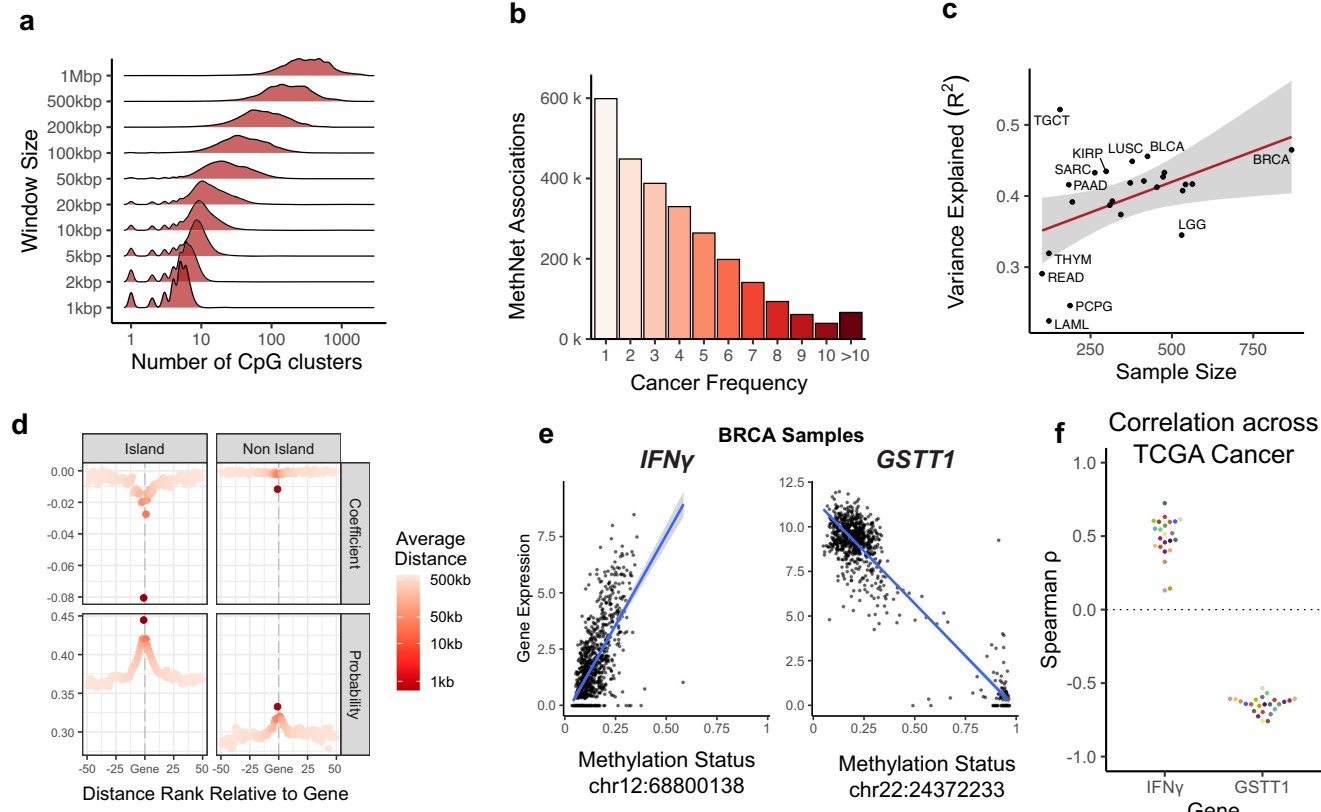

**Fig. 2 | MethNet identifies putative activating and repressing distal associations. a** Distribution of HumanMethylation450 probes neighboring a protein-coding gene as a function of the window size. At 1 Mbp the average gene has 400 potential regulators. **b** Histogram of the robustness of MethNet associations as measured by the number of TCGA cancers it appears in. **c** Performance of MethNet models, measured by the ratio of explained variance ($R^2$), as a function of dataset size. Trend line is fit with linear regression. Shaded area corresponds to 95% confidence interval of the mean performance given the number of samples in a cancer study ($n = 24$). **d** MethNet association effect as a function of distance. Associations are grouped based on their ranked distance to a gene, where −1 includes associations from the first CpG island or non-island upstream of the promoter, and positive distance refers to associations downstream of the TES. For every group of potential associations, we calculated the average distance, mean coefficient, and probability of a MethNet association. **e** Examples of regulatory effects recovered by MethNet in BRCA. Left: A repressive association between *IFNγ* and a CTCF binding site 250 kb upstream of the promoter. Right: An activating association between *GSTT1* and a non-coding RNA 10 kb downstream of the promoter. The linear regression line that is fit models the mean expression as a function of methylation status of the CRE. Shaded area corresponds to 95% confidence interval ($n = 868$). **f** Spearman correlation coefficient for the association shown in panel e across all TCGA cancers. We observe that the signal is robust across all cancers ($n = 24$).

the gene promoter. The potential of a CRE was calculated by summing all the distance-adjusted contributions to genes in its vicinity (Fig. 3a).

We next conducted a series of enrichment analyzes to uncover distinctive characteristics that are linked to a CRE's regulatory potential. First, we analyzed the chromatin state of each CRE using ChromHMM[32] (Fig. 3b). These investigations indicate that promoters acting as enhancers controlling other distal genes have the highest overall regulatory potential. Our observations are in line with the known regulatory role of distal gene promoters[9]. Indeed, the associations detected by MethNet transcend linear distance, highlighting its capacity to identify distal regulatory networks.

Methylation has been shown to directly affect the presence of the transcriptional machinery, chromatin modifiers, and binding of transcription factors (TFs) whose chromatin occupancy in turn, can act as a barrier to methylation[3,33–35]. To identify which factors could contribute to the regulatory potential of a CRE, we performed an enrichment analysis and identified members of the RNA polymerase complex (such as *POLR2A* and *POLR2G*) as the most enriched factors (Fig. 3c). Additionally, we identified other highly enriched factors known to be involved in chromatin remodeling and altering the methylation status, including *EGR1*, *HDAC1*, and *PHF8*. Depletion of EZH2 was also linked to increased regulatory potential which make sense as EZH2 is part of the

PRC2 complex which characterizes inactive chromatin. Moreover, using Hi-ChIP data from the benchmark study of Bhattacharyya et al.[36], we observed a strong positive correlation between the regulatory potential of a region and the number of active chromatin loops (H3K27ac loops) anchored at it, further highlighting the dynamic interplay between chromatin structure and regulatory activity (Fig. 3d).

## MethNet hubs that control multiple genes have an impact on patient survival

The distribution of the regulatory potential is long tailed, suggesting the existence of CREs with exponentially high regulatory potential (Fig. 4a) linked to multiple genes (Supplementary Fig. S4a, b). Intriguingly, these elements exhibited low methylation variance across the different TCGA datasets, indicating that they could be under robust and stringent regulation. We used the elbow method to show that above a select threshold, CREs were significantly enriched for regulatory associations. This approach identified 6137 CREs, that we refer to as 'MethNet hubs', since, as expected, their profile aligns with a model in which CREs exert control over multiple genes (Fig. 4b). A survival analysis, using the Cox proportional hazard model, across TCGA cancers with clinical data, revealed that methylation of MethNet

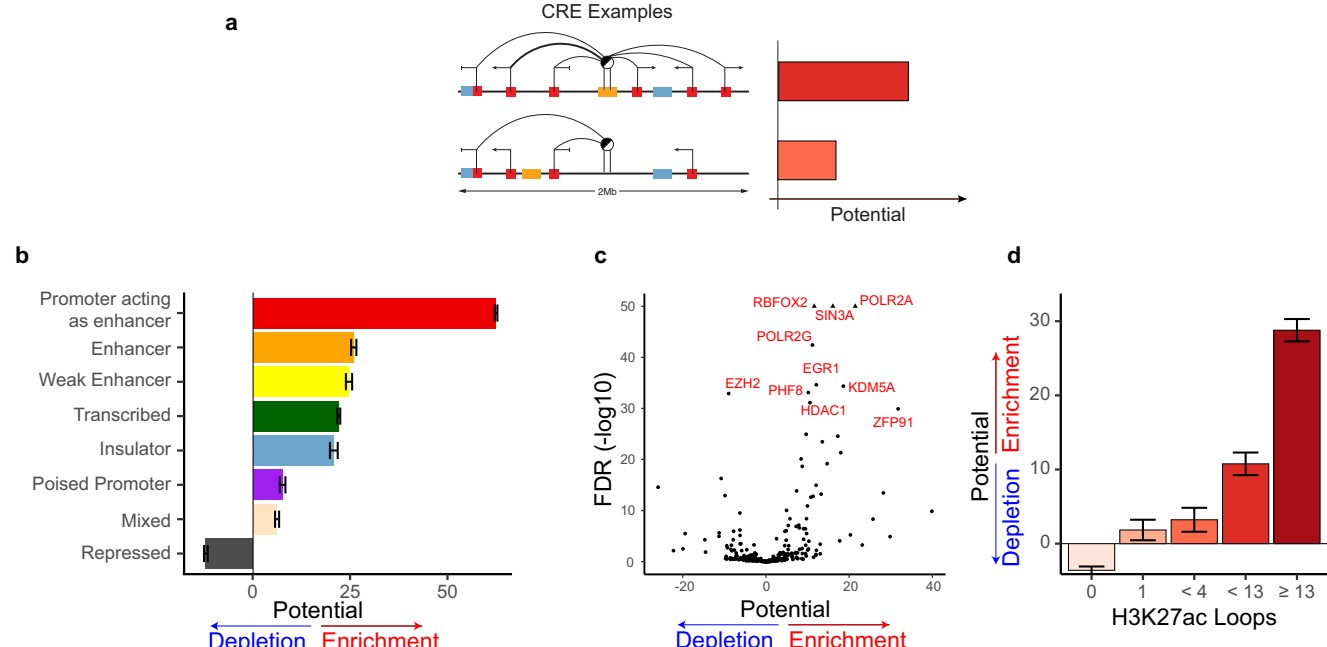

**Fig. 3 | The regulatory potential of CREs is correlated with chromatin context and contact frequency. a** Schematic depiction of regulatory potential. We quantified the importance of an association as the excess of its relative effect size with respect to a null model where all elements contribute proportionally to their distance from the promoter. **b** Enrichment or depletion of regulatory potential by ChromHMM state of the CREs (n = 245,511). Bar length represents mean effect compared to Low Signal state and error bars corresponds to ± the standard error (for details see Methods). **c** Enrichment or depletion of chromatin remodelers, the transcription machinery, and transcription factor binding sites. **d** Enrichment or depletion of H3K27ac chromatin loops from CD4-naïve T cells, GM12878 and K562 cells for CREs that do not overlap with protein-coding promoters (n = 166,552). Bar height represents mean effect on MethNet potential for each group compared to "0" loops which is the intercept. Error bars correspond to 95% confidence interval.

hubs has a bigger impact on the overall survival of patients in comparison to non-hub elements of similar variance, both in a pan-cancer and cancer-specific context (Fig. 4c and Supplementary Fig. S4c). This finding provides evidence for MethNet hubs having a pivotal role in the context of cancer biology and underscores their potential clinical relevance.

To gain insight into the characteristics of MethNet hubs, we repeated the previous enrichment analyzes, this time comparing hubs to other CREs with positive potential (Fig. 4d–f). As anticipated, we observed that hubs share many of the characteristics exhibited by high-ranking non-hub elements, however, key differences emerged upon closer examination. First, hubs are more likely located in open chromatin regions than non-hub CREs (Supplementary Fig. S4d, e). Next, in the ChromHMM and binding site analysis, insulating elements and CTCF were respectively enriched in MethNet hubs compared to non-hubs (Fig. 4d, e). This finding is consistent with the known role that insulating elements play in gene regulation via chromatin looping and insulated topologically associated domain (TAD) boundaries[37,38]. This hypothesis is further supported by our data showing that hubs are depleted in elements lacking chromatin loops (Fig. 4f). In summary, we identified MethNet hubs that are characterized by high-ranking regulatory potential and whose methylation status has low variance. Hubs are enriched for regulatory associations and insulating elements and show significant clinical relevance with respect to cancer patient survival.

## MethNet hubs uncover known and potentially novel regulatory elements

Examples of two regulatory hubs are shown in Fig. 5. The regulation of the Protocadherin alpha (*PCDHA*) cluster of genes has been shown to be controlled by a regulatory hub (HS5−1) that overlaps a CTCF binding site, which stochastically activates different *PCDHA* genes by cohesin-mediated looping[39,40]. Using MethNet we were able to unbiasedly

identify this regulatory hub (highlighted in blue). We also found another hitherto unknown regulatory hub that is enriched for H3K27ac and H3K4Me3 (highlighted in orange) upstream of *PCDHA*. This hub has predicted regulatory associations with genes from all three clusters of the Protocadherin family, *PCDHA*, *PCDHB* and *PCDHG*, and the region has been characterized as a schizophrenia risk locus that is linked with the regulation of all three protocadherin families in the context of brain[41]. In-situ Hi-C data reveals the high contact frequency of this CRE with all three Protocadherin families (as shown in red in the Hi-C heatmap).

## High scoring MethNet associations are mediated by long-range chromatin interactions

To determine whether distal CRE associations identified by MethNet are brought into contact with their target genes by chromatin looping, we performed a promoter-capture Hi-C experiment that identified chromatin interactions from all promoters in the genome in two distinct well characterized A549 and K562 cell lines. The quality control for the promoter-capture Hi-C is shown in Supplementary Fig. S5. Promoter-capture Hi-C, which enriches for loops anchored at gene promoters, allows us to simulate parallel 4C-seq experiments and recover regions that are in physical contact with each promoter. Given that MethNet consists of common and cell type specific associations, we would not expect all associations from our pan cancer analysis to be validated with the promoter-capture Hi-C data from the A549 and K562 cell lines. Indeed, an example of multi-locus interactions for the *TP53* gene promoter in A549 and K562 shown in Fig. 6a, highlights the cell type specific chromatin contacts found in the two cell lines.

To determine whether associations with higher scores are more likely to be facilitated via chromatin interactions, we used the union of loops from the A549 and K562 cell lines. We found a robust correlation between association score and probability of loop formation across all levels (Fig. 6b). This data demonstrates that stronger MethNet

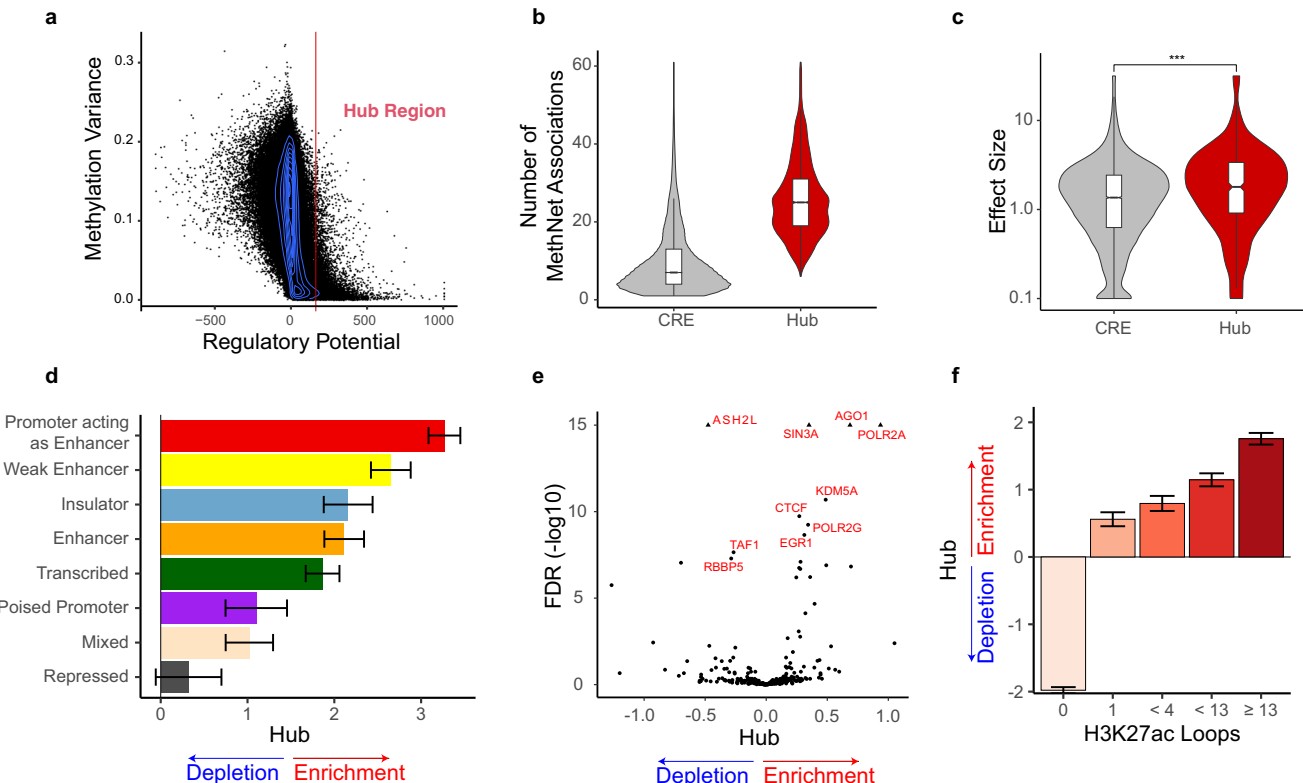

**Fig. 4 | MethNet hubs control multiple genes and have an impact on patient survival. a** Distribution of regulatory potential as a function of methylation variance across cancers. **b** Comparison of the distribution of MethNet association per elements for hubs (n = 6,139) versus non-hubs (n = 239,416). Boxplots were drawn with the following parameters: box bounds correspond to 1st and 3rd quantiles, center mark corresponds to median, whisker length is 1.5 the height of the box (inter-quantile region) or up to the extrema of the distribution if they are closer to the box bound. **c** Mean effect of hub methylation (excluding cancer-specific effects) on overall survival across TCGA cancers. A two-sided Wilcoxon rank sum test with continuity correction is used to calculate the $p$-value ($p$-value = $5 \times 10^{-11}$, $n_{Hub}$ = 574, $n_{CRE}$ = 174,139). Boxplots were drawn with default parameters as in panel

b. **d–f** Enrichment of regulatory potential hubs versus non-hub CREs with positive potential (see Methods for details). **d** Enrichment of hubs versus non-hubs CREs across ChromHMM (n = 121,517). Bar length correspond to mean effect of state versus Low Signal state and error bars correspond to 95% confidence interval. **e** Enrichment of hub versus non-hub CREs across chromatin remodelers and transcription factor binding sites **f** Enrichment of hub versus non-hub non-promoter CREs (n = 73,204) as a function H3K27ac chromatin connectivity from CD4-Naive, GM12878 and K562 cells. Bar height represents log odds (logit) effect size on the probability of a CRE being a hub as a function of its connectivity group, no loops is the intercept of the model. Error bars correspond to 95% confidence interval.

associations are more likely to act via chromatin contacts, while weaker connections may represent indirect effects.

Next, we investigated whether the regulatory potential of Meth-Net is predictive of chromatin hubs, i.e. CREs that form multi-locus loops[42,43]. Remarkably, MethNet's regulatory potential demonstrated a high predictive power for identifying multi-locus loops, achieving an area under the receiver operating characteristic curve (AUC) of 86%, when applying the strictest criteria (Fig. 6c). Although our analysis was primarily focused on promoter hubs (due to the experimental bias of promoter-capture Hi-C), the predictive power of MethNet extended to non-promoter regions when less stringent criteria were used for calling hubs (maximum AUC 84%, Supplementary Fig. S6). In sum, our data indicate that high-ranking distal MethNet associations are brought into close physical proximity by long-range chromatin interactions.

**Perturbation of MethNet hubs results in altered target gene expression**

To functionally validate the predictions generated by MethNet, we performed perturb-seq that combines targeted perturbation of genomic regions with single-cell RNA sequencing (scRNA-seq). Compared to a regular CRISPR interference (CRISPRi) assay, perturb-seq enables the simultaneous investigation of multiple genomic regions, using a pool of guide RNAs (sgRNAs). For the CRISPRi, we used the dCas9-KRAB-MeCP2 system that blocks the binding of other factors

and methylates the DNA as well as histones of the targeted region[44], inducing robust silencing. The transcriptomic readout aligns well with the MethNet pipeline, which predicts changes in the expression of target genes. Furthermore, perturbation of distal regulatory elements is more likely to lead to subtle gene expression changes rather than cell death[45], particularly when there is more than one CRE controlling a gene target. Supplementary Fig. S7 shows the percentage of A549 cells transfected with dCas9-KRAB-MeCP2 at day 14 after puromycin selection.

In total, we targeted 55 potential regulatory elements with 2 to 5 guides each. To address the inherent limitations of the assay, targets were selected based on the following criteria: (i) CREs were unme-thylated in A549 cells to allow for methylation by the dCas9-KRAB-MeCP2, (ii) 2 to 5 high-quality guides could be selected using the CRISPick[46,47] scoring system, and (iii) putative target-genes were expressed at levels detectable by scRNA-seq. An outline of the perturb-seq validation experiment is shown in Fig. 7a.

Although we designed our experiment so that each cell received a single guide, some cells contained multiple guides (Supplementary Fig. S8). Therefore, we used a linear model with a complex design matrix to deconvolve the individual effects of each guide on gene expression, similar to the approach used by Dixit et al.[48]. The results of this analysis are illustrated in Fig. 7b. Among the 55 targeted CREs, 17 were validated as being involved in 22 functional associations

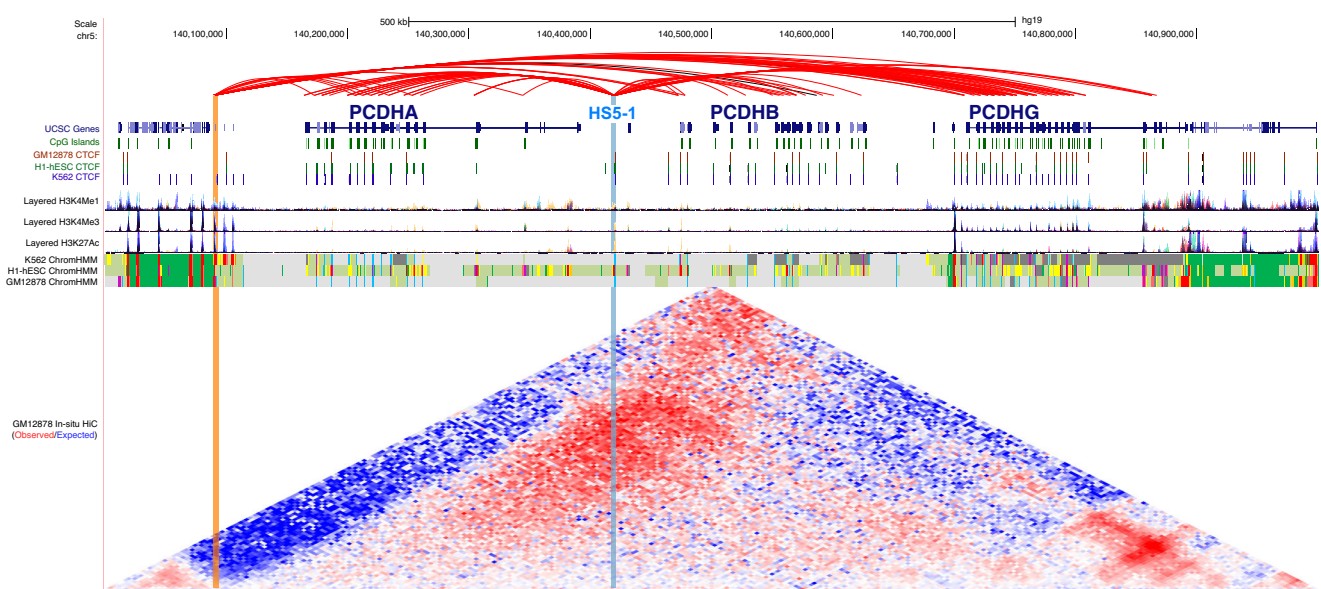

**Fig. 5 | MethNet hubs uncover known and potentially novel regulatory elements in the Protocadherin gene cluster.** (chr5:140,000,000–141,000,000) MethNet associations are shown in the top track. Red associations are activating and black repressing. All other tracks are from UCSC Genome Browser. Chromatin marks and CTCF binding sites data are provided by ENCODE. In-situ Hi-C data were generated by Rao et al., and processed with Juicebox to compute contact enrichment (colormap = $\log_2$(observed/expected normalized counts), range [−4, 4]). The HS5-1 enhancer, (highlighted in blue), is a known regulator of the *PCDHA* cluster. The enhancer on the left (highlighted in orange) is a MethNet discovery that is identified as a regulator of the *PCDHA, PCDH* and *PCDHA* clusters. The region of the previously unreported enhancer has increased contact frequency with all three Protocadherin families (highlighted by red in the Hi-C heatmap).

predicted by MethNet. To assess the significance of the identification of 17/55 functional CREs, we performed a bootstrap analysis, by randomly shuffling the sgRNA labels across cells, while maintaining the total number of detected guides. We generated 20,000 bootstrap samples to estimate the null distribution (Fig. 7c) and estimated that the probability of detecting 17 or more regulatory regions was highly unlikely to have arisen by chance (p = 0.0004). This statistical evaluation supports the robustness and significance of the regulatory regions identified through perturb-seq.

Out of the 17 functional CREs identified, 4 were found to be associated with more than one target gene. In Fig. 7d we highlight a regulatory hub that corresponds to the promoter of *BNIP2* (highlighted in orange). *BNIP2* itself is not highly expressed so it was not captured by the perturb-seq, but expression of two predicted target genes, *GCNT3* and *ANXA2* (red loops) were found to be significantly downregulated. Both of these genes are associated with poor prognosis in multiple cancers. *ANXA2* is a member of the calcium-mediated phospholipid-binding protein family of annexins, involved in epithelial mesenchymal transition, cell proliferation and survival[49], while *GCNT3*, is a member of the N-acetylglucosaminyltransferase family that is associated with cell proliferation, migration and invasion in non-small-cell lung cancer[50]. Our data indicate that the hub corresponds to a promoter region that is predicted to act as an enhancer for *GCNT3* and *ANXA2*, and thus methylation leads to a drop in their expression as we observed. We also identified two genes, *MYO1E* and *LDHAL6B* that are predicted by MethNet to be targets of the hub (highlighted by gray loops in the screenshot) that we were unable to validate. Although both gene targets are expressed and unmethylated in A549 cells, unlike *ANXA2* and *GCNT3* they were not connected to *BNIP2* by chromatin loops in our promoter-capture Hi-C. Chromatin looping is likely to be important for long-range hub mediated regulation, and we speculate that in the case of these two target genes, contacts are regulated in a cell type specific manner. Other MethNet associations validated by the perturb-seq experiment are shown in Supplementary Fig. S9. These include the target gene *AMIGO2*, a cell adhesion protein that is linked with cell survival and metastasis of multiple adenocarcinomas[51,52]. *GFBP4*, a tumor suppressor acting as double-negative feedback in *AKT* and *EZH2* signaling[53] (Supplementary Fig. S10). Taken together, these data confirm the robustness of the MethNet pipeline in predicting regulatory associations.

## Discussion

Here we introduce MethNet, a pipeline that combines gene expression and methylation data from the same TCGA cancer samples to identify regulatory elements that can control genes beyond their immediate genomic vicinity. MethNet's unbiased approach led to the identification of numerous regulatory features commonly associated with the role of methylation in gene expression. In addition, it revealed the existence of previously unknown regulatory elements with potential clinical significance. MethNet's most intriguing attribute is the ability to uncover the presence of regulatory hubs that can influence the expression of multiple genes. These hubs displayed the expected characteristics of previously identified hubs[10], such as enrichment in active chromatin marks and chromatin looping connections between hub CREs and their target genes. MethNet also revealed hub CREs, including a relatively high proportion of insulator elements compared to regular CREs and low methylation variance, suggesting that hub regulation is influenced by TAD structure and that they are tightly regulated. Moreover, the methylation status of hubs showed a significant correlation with overall patient survival as well as cancer-specific survival.

It should be noted that our modeling of epigenetic context was limited by the availability of data. The DNA methylation profiles provided by TCGA were generated using the 450k array, whereas larger 850k arrays are now widely used. Furthermore, TCGA lacks other relevant data modalities, such as chromatin accessibility and protein binding profiles, that are important for a more complete understanding of the regulatory landscape. As the volume and diversity of genomic data expand, MethNet can be adapted to incorporate additional data modalities, further enhancing its capacity to unravel the complexities of gene regulation.

Any method that identifies regulatory elements by linking gene expression with methylation must deal with the problem of spurious correlations. This issue is exacerbated by the fact that we consider long-range (up to 1 Mbp) associations and that nearby methylation

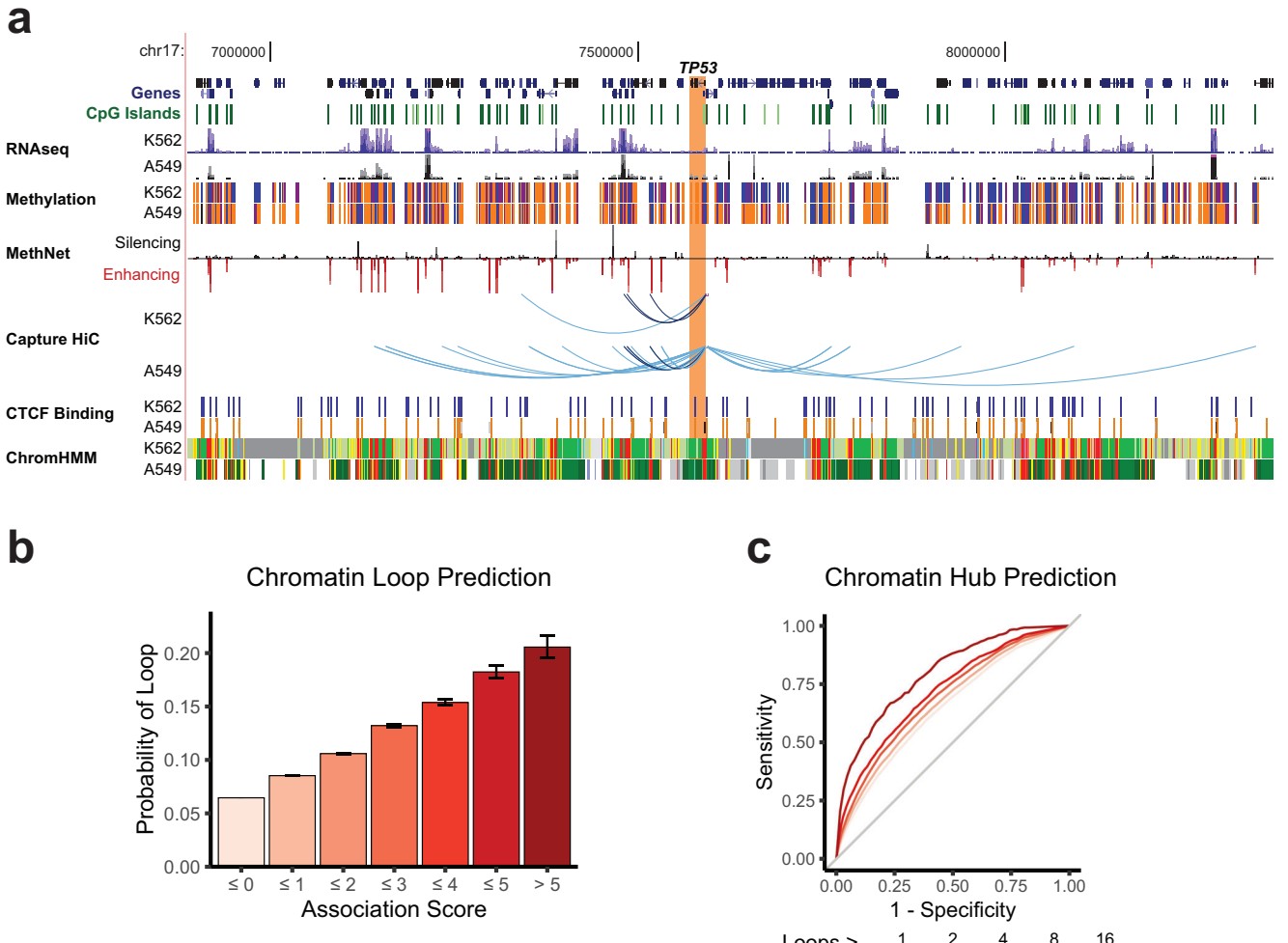

**Fig. 6 | High scoring MethNet associations are mediated by long-range chromatin interactions. a** Example of MethNet associations regulating *TP53* that overlap chromatin loops identified using promoter-capture Hi-C from the A549 and K562 cell lines. The Genome Browser session shows the chromatin context around the promoter of *TP53* (chr17:6,886,465-8,364,371) for both cell lines. This includes RNA-seq (ENCODE/Caltech for K562, ENCODE/HAIB for A549 ETOH), methylation status (ENCODE/HAIB - orange and blue correspond to methylated and unmethylated regions, respectively), CTCF binding sites, and ChromHMM chromatin states for K562. A549 ChromHMM states were downloaded from Roadmap Epigenomics (15-state core model). All tracks were loaded with default settings, except RNA-seq which was capped at the top for a better overview. Red and black bars correspond to predicted activating and repressive MethNet associations, respectively. Promoter-capture Hi-C loops (shown as arcs) that overlap with MethNet CRE predictions are shown in virtual 4 C format. Darker arcs correspond to loops called in both cell lines. **b** Bar graph depicting the probability of a MethNet association overlapping with a chromatin loop in either cell line as a function of its score ($n = 1,585,070$). Bar heights correspond to the probability of an association of the corresponding group overlapping a loop computed using a logistic regression model. Error bars correspond to the 95% confidence interval. **c** ROC curves showing the ability of MethNet potential to predict chromatin hubs. We only considered gene promoters because of experimental bias. The AUC increases for stricter criteria of hub calling.

sites tend to be correlated and can act synergistically. To address the confounding effect of correlated methylation sites, we clustered probes within 200 bp into a single variable. Clustering neighboring CpG sites is standard procedure for smoothing technical noise and reducing biological artifacts, such as genetic variants that destroy CpG sites. Clustering increases the power of the analysis without losing information, since methylation of proximal CpGs is highly correlated. In addition, it is well documented that methylation changes are found in differentially methylated regions (DMR) typically spanning ~100–1000 bp regions. The 200 bp window is a standard size for CpG clustering, as it encompasses both typical transcription factor binding sites and the length occupied by histones. We evaluated the performance of the clustering using three metrics: mean cluster size, number of clusters and coefficient of variation (Supplementary Fig. S10).

Although 1Mbp range interactions are important for gene regulation, testing all possible promoter-CRE pairs is prone to produce high false discovery rates. We addressed this issue by combining data across genes and cancers in a statistically principled manner. In particular, we used elastic-net regression, tuned with cross-validation, to promote sparsity within every cancer and then pooled the resulting associations across cancers based on their predictive strength, while accounting for known confounding factors (see Methods: MethNet score). This multi-level approach significantly reduced the number of identified CREs per gene compared to a naïve analysis of variance with lasso-penalty (Supplementary Fig. S11).

Previously developed methods try to mitigate the problem of spurious correlations by limiting the range of associations and by using permutation or cross-validation techniques. Our modeling approach is similar to that of Methylation-eQTL in that it also uses TCGA data and penalized regression to identify regulatory associations. Both methods assume a linear additive model, which is a pragmatic choice given the size of the currently available data set.

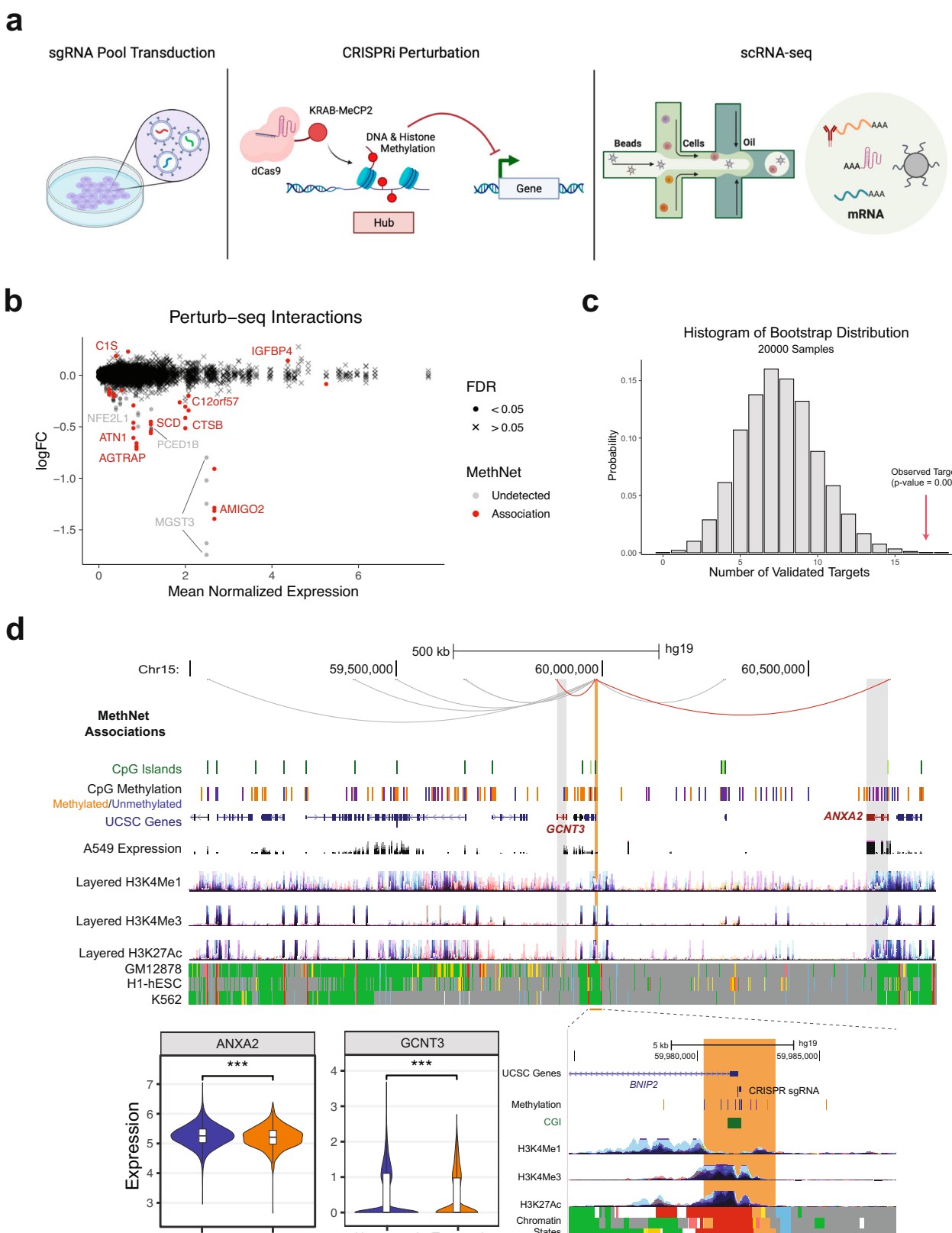

However, while Methylation-eQTL limits its scope to a 500 kbp window around the gene and uses a sequential lasso approach to deal with promoter sparsity in individual tumors, we analyzed a 1 Mbp window around each gene and employed elastic-net regularization, which leads to a less sparse solution and combats spurious CRE associations by aggregating results across multiple cancers. Another

important difference between the two approaches is that MethNet, in addition to focusing on individual CRE-promoter connections, uncovers regulatory hubs and highlights their relevance in the context of normal gene regulation and cancer. In contrast to Methylation-eQTL, which only performed cross-validation using an independent data set, our experiments provide causal and

**Fig. 7 | Perturbation of MethNet hubs results in altered target gene expression. a** Outline of the perturb-seq validation experiment. **b** MA plot showing the differential expression induced by CRISPRi targeting of regulatory regions. Points correspond to genes targeted by sgRNAs. Point color indicates whether the interaction was predicted by MethNet. Point shape indicates whether the interaction was considered significant using the False Discovery Rate (FDR) of 0.05. **c** Bootstrap distribution of the number of targeted regions that show significant changes in gene expression. The number of observed validated targets is marked by a red arrow. **d** Genome Browser session showing an example of a CRE hub validated by expression changes in two predicted gene targets. A zoomed in version of the browser around the hub regulatory element is shown. The effect of all sgRNA

guides targeting the CRE on expression (log of normalized counts) of target genes across all cells is shown in the violin plots and boxplots ($n_{Untargeted} = 17,790$, $n_{Targeted} = 1301$). Boxplots were drawn with the following parameters: box bounds correspond to $1^{st}$ and $3^{rd}$ quantiles, center mark corresponds to median, whisker length is 1.5 the height of the box (inter-quantile region) or up to the extrema of the distribution if they are closer to the box bound. A Welch two-sided t-test is used to calculate the $p$-value ($p_{ANXA2} = 2.8 \times 10^{-6}$, $CI_{ANXA2} = [0.028, 0.069]$, $p_{GCNT3} = 1.3 \times 10^{-4}$, $CI_{GCNT3} = [0.034, 0.106]$). Figure 7/panel a Created with BioRender.com released under a Creative Commons Attribution-NonCommercial-NoDerivs 4.0 International license (https://creativecommons.org/licenses/by-nc-nd/4.0/deed.en).

mechanistic support for our findings, by validating the functional and physical connections between hub CREs and their target genes.

Our motivation in this study, was to uncover interactions that are robust across multiple cancers rather than cancer-defining CRE-gene interactions. There have been several studies that identify tissue-specific methylation[54,55] and gene markers[56]. MethNet does not aim to recover all cancer-specific regulatory associations but instead identifies a core, robust set of dynamic associations that are recurrent across tissues and revealed by cancer deregulation. These core CREs are likely to be important for carcinogenic processes in general, regardless of the tissue type, and they could be useful for understanding mechanisms underlying cancer and for identifying targetable hotspots. We, therefore, treated each TCGA-cancer study independently and normalized the coefficients to produce dimensionless units that are comparable across cancers. Cancer-defining CRE-gene interactions were detected as low confidence associations which were effectively ignored because our focus was to identify robust cross-cancer CREs.

MethNet identified 37,740 distal regulatory elements, including the regulatory hub in the *PCDHA* cluster. This hub mapped to a GWAS (genome-wide association study) schizophrenia risk locus, supporting the functional relevance of the pipeline. While GWAS studies have identified more than 100,000 disease-associated SNPs (single nucleotide polymorphism) over the past decades[57], the identification and prioritization of causal germline SNPs remains challenging due to linkage disequilibrium and the size of the non-coding genome. Nonetheless, identifying causal SNPs is of paramount importance in understanding the underlying mechanisms of disease susceptibility and leveraging GWAS data. Towards this end, MethNet provides a valuable resource for prioritizing the verification of predicted non-coding disease variants that are likely to disrupt regulatory elements versus the numerous 'proxy' variants in high linkage disequilibrium that share the same disease-association statistics, but have no functional effect.

The results from the Pan-Cancer Analysis of Whole Genomes (PCAWG) and TCGA consortiums, which include more than 2500 cancer samples, were somewhat disappointing as only a few non-coding somatic driver mutations were identified[58]. This can be partly explained by the fact that the statistical approaches to identify coding driver mutations are not tailored to the non-coding genome. Furthermore, it has been shown that methods based on functional screening followed by experimental validation are better able to identify bona fide non-coding driver mutations[59], compared to approaches that focus on the accumulation/recurrence of non-coding mutations. In this context, MethNet, in addition to being able to prioritize candidate non-coding driver mutations, can also identify the target genes of disrupted regulatory elements, which is crucial for developing novel therapeutic drugs.

Many studies have pointed out that regulation is more complex than simple enhancer-promoter loops[9]. For example, super-enhancers (SEs), which are large regions (around 8 kb) characterized by strong enrichment in chromatin marks like Med1 or H3K27ac[60], are associated with highly expressed, tissue-specific genes[61]. In addition, there are CREs with multi-locus contacts that influence the expression of numerous genes. Our analysis of the consensus regulatory network

revealed the existence of both individual CRE-promoter contacts, as well as MethNet hubs that have an outsized influence on gene expression.

In this study, we took advantage of a high-throughput perturb-seq assay that combines CRISPRi screening with scRNA. This assay is well suited to the functional screening of distal regulatory elements since it allows the identification of relatively modest gene expression changes upon perturbation of these elements. While coding mutations or disruption of gene promoters can lead to dramatic changes in cell fitness, it has been shown that the regulatory potential of distal elements might be more subtle, although these are likely to be important in disease processes.

Using perturb-seq, we were able to demonstrate that about one-third of the targeted regulatory hubs were associated with transcriptional changes in cancer-related genes, suggesting that a substantial proportion of MethNet hubs contribute to the cancer cell phenotype by activating or silencing oncogenic and tumor-suppressor genes, respectively. In addition, the validated MethNet associations were not trivial, but linked genes to regulatory elements over long distances (mean distance 366 kbp) bypassing multiple potential regulatory elements in between. Among these validated MethNet hubs, we observed expression changes in at least one of the predicted target genes, while in others no alterations were detected. This could be because: (1) Gene expression levels could be below the threshold detectable by scRNA-seq, and (2) MethNet hubs are defined using a pan-cancer approach, and while this allows us to prioritize the most robust candidates, it does not rule out context-specific gene regulation, as we highlight in Fig. 7, where predicted hub-target gene associations did not overlap with chromatin loops in the cell line analyzed. Therefore, although MethNet is useful in identifying high-confidence regulatory hubs, studying their gene regulation network in a tissue-specific manner is also important.

In conclusion, MethNet represents a powerful computational framework for the integrative analysis of DNA methylation and gene expression data. Our study demonstrates the effectiveness of MethNet in identifying regulatory associations across multiple cancer types as well as context-specific connections, highlighting the importance of functional integrative analysis over simple correlations. The performance of MethNet scales with data, indicating its potential for further expansion with larger datasets and inclusion of other data modalities. Identification of previously unreported hubs and their association with clinical outcomes sheds light on the intricate interplay between chromatin structure and global transcriptional regulation. Overall, MethNet represents a valuable resource for deciphering the complex regulatory mechanisms underlying gene expression in cancer, and for prioritizing the validation of germline and somatic non-coding disease-associated variants.

## Methods

### Promoter Capture HiC sample and library preparation
Promoter Capture Hi-C data was generated in K562 and A549 cell lines using the Arima Capture-HiC+ Kit (catalog number: A301010, including, the Arima Promoter Capture Module, and the Arima Library Prep Module according to the Arima Genomics manufacturer's protocols.

The A549 and K562 cell lines were purchased from ATCC (catalog number: CCL−185 and CRL-3343, respectively). Two replicates of the Hi-C were performed in each cell line, and for each replicate 1 million cells were collected and double cross-linked using 3 mM DSG (disuccinimidyl glutarate), followed by 1% formaldehyde. Samples were sequenced with Novaseq Illumina technology according to standard protocols with around 300 million (150 bp paired-ends) reads per sample. The library preparation and sequencing were conducted by NYU Langone's Genome Technology Center.

## Perturb-seq sample and library preparation

CloneTracker XP CRISPR Barcode pooled lentiviral libraries expressing barcoded sgRNAs, the puromycin selection gene as well as an RFP reporter were purchased from Cellecta® (Catalog number: custom library, CPLVSGL-P; lentiviral packaging service, CLVP-V). The plasmid (pGC02-EFS-KRAB-dCas9-MeCP2-2A-Blast) was a gift from Dr. Neville Sanjana. The A549 cells were transduced with lentiviruses expressing dCas9-KRAB-MECP plasmids as described in Yeo et al.[42]. Cells were then grown in DMEM medium (Gibco/Invitrogen) +10% FBS +100 units/ml penicillin +100 μg/ml streptomycin +5% CO2 at 37 °C.

In total, we targeted 55 potential regulatory elements with 2 to 5 guides each. To address the inherent limitations of the assay, targets were selected based on the following criteria: (i) CREs were unmethylated in A549 cells to allow for methylation by the dCas9-KRAB-MeCP2, (ii) 2 to 5 high-quality guides could be selected using the CRISPick[44,45] scoring system, and (iii) putative target-genes were expressed at levels that were detectable by scRNA-seq. An association was considered detectable if the gene was highly expressed and MethNet predicted that methylation would lead to silencing or when the gene is lowly expressed and MethNet predicts that methylation leads to expression. We also included 3 partially methylated CREs and 2 with no detectable targets in A549 as controls (Supplementary Data 1: **Perturb-seq design** – Sheets for CRE and Gene criteria).

The 248 sgRNAs targeting high-confidence MethNet hubs and 5 non-targeting sgRNAs (negative controls) were designed using CRISPick[44,45] for the Human GRCh37 (hg19) assembly using parameter settings: CRISPRi, SpyoCas9/Chen (2013) tracrRNA (Supplementary Data 1: **Perturb-seq design** – Sheet for sgRNA Sequences). The lentiviral libraries were transduced into the A549-dCAS9-KRAB-MeCP cells according to Cellecta® protocols. Briefly, $10^5$ cells/well were seeded into a 6 well-plate. The optimal MOI of viral particles (to reach ~30–40% of infected cells) was added. On day 3, 2 ug/ml of puromycin was added, resulting in >90% transduced cell selection as confirmed by cytofluorometry using RFP (Supplementary Fig. S7). Cells were expanded under puromycin selection until day 14. For scRNA-seq using the 10X Genomics technology, 25,000 cells were harvested. For optimal multiplet detection and optimal signal-to-noise ratios, cells were hash-tagged using 5 cell multiplexing oligos purchased from 10X Genomics (catalog number: 1000261, 1000262, 1000243, 1000242). The sequencing library was, then, prepared using the Chromium Next GEM Single Cell 3′ Kit (catalog number: 1000268, 1000120 and 1000215) according to the manufacturer's protocol and sequenced by NYU Langone's Genome Technology Center.

## Statistics and reproducibility

No samples were excluded from the analyzes, unless they contained missing data. We limited our analysis to protein coding genes and cancer studies with at least 100 samples with matching RNA-seq and DNA methylation data. In total, we used 8,264 samples. We used GENCODE v30 gene annotation[62] and the Illumina CpG probe coordinates for hg19.

## Gene expression modeling

We constructed a gene-probe adjacency network by connecting a gene with all probes within 1Mbp on either side of its transcription start site (TSS). This resulted in a network of 13 M interactions between 20k genes and 450k probes. We collapsed probes that were within a diameter of 200 bp (complete linkage) into probe clusters by averaging their beta value and linking them with all the genes interacting with the original probe set. This resulted in 300k CpG clusters, of a single probe or more, and a new network of 8 M interactions.

We fitted a linear model for each gene and TCGA cancer independently, removing gene-cancer pairs with low variance (standard deviation less than 1). The variables of the model included all the methylation clusters neighboring the gene in the adjacency network as well as the sample type (tumor or metastatic vs normal) wherever there were multiple sources. To promote sparsity and better generalization in our models we used elastic net regularization to limit the number and effect size of the cluster-gene interactions using the model specification:

$$y_{gi} = \beta_{g0}(z_i) + \sum_c^{k_g} \beta_{gc} x_{ic} + \epsilon_{gi}$$
$$\min_\beta \epsilon_{gi}^2 + \lambda_g \left( \alpha_g |\beta_{gc}| + \left(1 - \alpha_g\right) \beta_{gc}^2 \right)$$

(1)

Here $i$, $g$ and $c$ are indexes for the sample, gene, and cluster, respectively. $y_{gi}$ is the log-normalized expression of $g$ in sample $i$, $\beta_{g0}$ is the basal level expression for sample $i$ given its clinical profile $z_i$ (tumor or normal), $\beta_{gc}$ and $x_{ic}$ are the coefficient and methylation status (beta value) of cluster $c$ respectively, and $k_g$ is the number of clusters neighboring $g$. $\epsilon_{gi}$ is the error term of the model. The R package `glmnet`[63] was used to fit the models and determine the trade-off $(\lambda, \alpha)$ between accuracy and sparsity via 10-fold cross validation. This process resulted in a series of regulatory networks, one per cancer, connecting genes with methylation clusters if in the corresponding gene model, the cluster had a non-zero coefficient.

For our pan-cancer analysis, we combined the interaction coefficients across all cancers by averaging across all cancers weighted by the corresponding model's performance as measured by $R^2$.

## MethNet score

To quantify the contribution of each gene-cluster interaction to overall gene expression, we calculated a contribution score based on the following function:

$$b_{gc} = \log \frac{|\beta_{gc}|}{\sum_c |\beta_{gc}|} - \log \frac{1}{k_g}$$

(2)

In contrast to the coefficient $(\beta_{gc})$, which quantifies how much the expression of gene $g$ is altered if cluster $c$ switches from a completely unmethylated to a completely methylated state, the score $(b_{gc})$ intends to capture the information gained by using MethNet's regulatory network instead of a naive model where all neighboring elements contribute equally $(1/k_g)$. Scores were computed based on the consensus network (pan-cancer analysis).

Finally, to identify the intrinsic potential of a regulatory element, we regressed out the effect of the distance $(d_{gc})$, which acts in an element agnostic manner, from the MethNet score using a GAM model with the absolute distance between TSS and methylation cluster as the only predictor, and taking the residuals. Thus, for the purposes of characterizing regulatory hubs we assume a distance-based, instead of an equivalence contribution of each region. To fit the GAM model $f(d_{gc})$, we excluded associations that overlap the gene body or the promoter since they are mediated via different mechanism of action and can would confound the effect.

The MethNet potential of a cluster is defined as the sum of scores of all the interactions it's part of:

$$b_c = \sum_g (b_{gc} - f(d_{gc})) \qquad (3)$$

In total, we computed the MethNet potential for 245,555 CRE candidates. MethNet association and CRE scores are provided as supplementary data methnet.csv and cluster_score.csv at figshare (see Data Availability).

## Regulatory effect as a function of distance

We analyzed the relationship between MethNet associations and distance to gene separately for CpG sites located within and outside the gene body.

For interactions occurring outside the gene body, we aggregated CpG clusters into two categories: CpG island and non-island regions. Subsequently, for each gene, we separately ranked these categories based on their distance to the gene body, where −1 is the closest upstream region, −2 the second closest etc, and accordingly 1, 2 for upstream regions. Two metrics were used: the mean coefficient of interaction and the probability of interaction, which were calculated for all clusters within the respective region.

## Enrichment of regulatory potential

To assess the enrichment of regulatory potential, we performed an overlap analysis between CpG clusters and ChromHMM states. A linear model was fitted using all 245511 CREs, with the Low Signal state serving as the reference baseline. In this context, the enrichment score is the difference between the average regulatory potential of clusters overlapping a ChromHMM state versus the Low Signal state. Similarly, we conducted a similar analysis for transcription factor binding sites, allowing for the possibility of multiple factors binding to a single site to account for confounding effects. In this case, the basal state represents unbound chromatin, and the enrichment score is the average difference in regulatory potential of clusters bound by a specific TF versus those that are unbound.

The resolution of the H3K27ac loops was 2.5 kb. We overlapped anchors with CRE candidates and filtered out CREs that overlap promoters, defined to be with 2000bp of any TSS, to focus on enhancer elements. In total, we analyzed 166,552 CRE candidates grouped into 5 groups based on the number of loops: 0, 1, [2, 4), [4, 13), [13, 209). A linear regression model was fit to estimate the enrichment score.

To assess the enrichment of hubness, we repeated the same process but instead of a linear model, we fitted a logistic regression model to predict hub vs non-hub among all the regulatory elements with positive regulatory potential.

## Survival analysis

We excluded from this analysis elements with low methylation variance (bottom 25%) and from the rest we selected all the hubs (730) and 16190 at random. A Cox proportional hazard model on overall survival (OS) was fitted for each element using the survival R package[64]. The methylation status (beta) was used as a predictor and we fitted a varying-slope model: coxph(Surv(OS.time, OS) ~ beta*strata(cancer), to estimate both the mean effect of methylation across all cancers (Fig. 4c) and the cancer specific effect (Supplementary Fig. S4c for coefficients with less than adjusted p-value 0.05).

## Analysis of promoter capture HiC

We called loops using the Arima pipeline [https://github.com/ArimaGenomics/CHiC] with default parameters: which is based on HiCUP[65] and CHiCAGO[66]. We ran the analysis independently for the K562 and A549 cells and then used the union of loops for subsequent investigations. Quality control metrics and loop replication are shown in Supplementary Fig. S5.

The resolution of loop anchors was 5 kb, so we only considered associations of length 10 kb and above. We considered an association overlapping a loop if loop anchors overlapped with both the regulatory element and the gene. Enrichment was computed on the basis of associations: associations were grouped into bins and a logistic regression was used to compute their probability to overlap with a called loop.

To estimate the predictive power of the MethNet potential to call hubs, we first computed the number of loops anchored at each bin and then we computed the potential of the bin by summing the potential of all the clusters contained in it. Finally, we called hubs for different thresholds and estimated the predictive power of the bin's potential using the AUC of the ROC curve using pROC packages.

## Analysis of perturb-seq

Cells were called and analyzed using the *10x Genomics Cell Ranger 7.0.0* for initial cell calling and protospacer calling. The results were further filtered based on the percentage of mitochondrial reads and total number of genes, to remove lysed cells, and based on the HTO tags to remove duplicates using the hashedDrops function of the DropletUtils package[67] and manually filtering to remove what appeared to be triplets per drop. After these filtering steps we ended up with 36,601 cells. The distribution of the number of expressed genes and number of reads per cell as well as the number of guides per cell are shown in Supplementary Fig. S8.

Subsequent analysis was based on the method suggested by Dixit et al.[48]. In particular, we focused only on genes that were within 1 Mb of a targeted region (as this is the maximum radius of MethNet association) and genes with a mean normalized expression above 0.0002. Since most cells had more than one sgRNA guide we analyzed the results for all the targets simultaneously. In particular, we identified sgRNA-gene interactions by fitting a linear model $log(Y) \sim X\beta$, where $log(Y)$ are the log-normalized counts (logNormCounts) and $X$ is a binary matrix where $X_{ij} = 1$ if the guide j is detected in the cell i. The models were fitted using the limma package[68] with Bayesian shrinkage, and interactions were called using a threshold of 0.05 on the adjusted P-value. All recovered associations are shown in Supplementary Fig. S9. For the bootstrap analysis, we shuffled the rows of $X$ independently for each column and refitted the model. The metric we used to quantify the performance of MethNet was the number of targeted regions forming at least a single interaction at the 0.05 threshold. Since the confidence interval is affected by the number of cells transfected with each guide, this process controls for spurious interactions due to the differences in the cell base of each guide.

## Reporting summary

Further information on research design is available in the Nature Portfolio Reporting Summary linked to this article.

# Data availability

The results published here are in part based upon data generated by the TCGA Research Network [https://www.cancer.gov/tcga]. We collected paired gene expression and methylation data via the xenahubs portal[69] which downloaded data from gdc.cancer.gov (data release 9.0 - October 24, 2017). We used the pan-cancer batch-corrected normalized gene expression [https://www.synapse.org/#!Synapse:syn4976369] and beta values for methylation from Illumina's Human-Methylation450 BeadChip [https://www.synapse.org/#!Synapse:syn4557906]. Clinical data for those samples was downloaded from xenahubs where available [https://www.synapse.org/#!Synapse:syn8402823]. Metadata for the CpG probes were collected from Illumina's annotation of HumanMethhylation450 BeadChip via the IlluminaHumanMethylation450kanno.ilmn12.hg19 Bioconductor package. The annotation was augmented (using the custom script

annotate_clusters.R) by overlapping clusters with tracks from the UCSC genome browser[70]. We used the ChromHMM chromatin annotation, CG island, and the transcription factor binding site cluster tracks. Links for all the data downloaded for this annotation are included in the custom script. Annotations were based on the most common labeling across all cell types. DNAse data for K562 cells were downloaded from ENCODE, we used the DNase regions of the combined replicates (file id ENCFF621ZJY). Hi-ChIP loops were downloaded from the supplementary material of the FitHiChIP paper[34]. We used the combined loose and merged replicate (L + M) loops for 2.5 kb bins for all cell lines (CD4-Naive, GM12878, K562). The primary data for Hi-ChIP loops were generated by Mumbach MR et al.[71] [https://www.ncbi.nlm.nih.gov/geo/query/acc.cgi?acc=GSE101498]. The manually processed data are shared at the GitHub repository of MethNet (see Code Availability) as Bhattacharyya_loops.csv.gz. The raw and processed sequencing data generated in this study have been submitted to the Gene Expression Omnibus (GEO) database under the superfamily accession number GSE236305. The promoter capture Hi-C and Perturb-seq data accession numbers are GSE235851 and GSE236304, respectively. The results of MethNet analysis used to generate the figures are uploaded to figshare [https://doi.org/10.6084/m9.figshare.25988074.v3]. No previously published data are under restricted access. Source data are provided with this paper.

## Code availability

The code used to generate the figures is available at https://github.com/TeoSakel/MethNet. Zenodo https://doi.org/10.5281/zenodo.11404065.

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

## Acknowledgements

These studies were supported by a P01CA229086 (JAS, AT, AH) and 2R35GM122515 (JAS). GC and GJ were supported by fellowships from the NCC.

## Author contributions

These studies were designed by Theodore Sakellaropoulos, Jane A Skok, Aristotelis Tsirigos and Catherine Do. All the analysis was performed by Theodore Sakellaropoulos. The Perturb-seq experiment was performed by Guimei Jiang; the promoter-capture Hi-C by Giulia Cova, Sitharam Ramaswami, and Dacia Dimartino, supervised by Adriana Heguy. scRNA-seq for the perturb-seq was performed by Peter Meyn. The paper was written by Theodore Sakellaropoulos, Catherine Do and Jane Skok.

## Competing interests

Aristotelis Tsirigos is a scientific advisor to Intelligencia AI. The rest of the authors declare no competing interests.
