## [Peer Review File · Nature Communications]

Editorial Note: Figure 1 and 2 on page 37 of this Peer Review File have been redacted as indicated to remove third-party material and to maintain confidentiality where no permission to publish could be obtained.

REVIEWER COMMENTS

Reviewer #1 (Remarks to the Author): Expert in gene regulatory networks, epigenetics, multi-omics, computational systems biology, and cancer omics

The manuscript by Theodore et al. introduces the method MethNet for identifications of interactions between cis regulatory elements and the gene promoters. MethNet takes advantages of the DNA methylation and gene expression data in large cancer cohorts. The authors then presented CRE hubs that appear to regulate multiple genes. High-throughput experiments then confirmed the physical interactions between CREs and the target genes. Functional assays were also performed to explore the biological relevance of the CREs. However, I have some major concerns regarding the novelty, the biological relevance, and the methodology design of this work.

1. The analysis between CpG probes and genes has been arbitrarily limited within 1 Mbp around the TSSs. Why? What is the molecular basis of such setting?
2. The authors mentioned that the data of CpG probes within 200bp were collapsed simply by taking averages of the beta values. I don't think this is appropriate. Why 200 but not 100 or 300? Did the authors used a sliding window of 200 bp to scan the whole genome, or did they simply partition the genome into regions of 200bp?
3. A simple linear regression model was used to evaluate potential regulation between a CpG cluster and a gene. I don't see novelty of this method design. This model is also too naïve, and the field has long recognized that such simple models can barely recapitulate the complexity of gene regulation programs.
4. Finally, a pan-cancer model was built by taking averages, again, of the coefficients obtained across multiple cancer types. Such an operation simply wiped out the cancer heterogeneity, and this is not a s-called pan-cancer network.
5. MethNet uses an average of 400,000 (5%) associations per individual cancer to model the expression of every gene.
6. Instead of combining all the data points from all the cancer types, Fig. 2e and 2f should show the positive and negative associations in one cancer type as an example. The associations in different cancer types could be summarized as box plots, for example.
7. As shown in Fig. 2a, there are hundreds of CpG clusters around each gene. This would for sure result in high false-positives. Of course many known cis-regulation would be recovered by MethNet, but this does not address the issue of high false-positive rates.
8. Fig. 3b, what does "promoter of other genes" mean? In the main text, the authors stated this term as "promoters acting as enhancers controlling other distal genes".
9. How were the numbers of "potential enrichment" calculated in Fig. 3? Do the negative values mean depletion? The authors reported positive and negative enrichments all together in the main text (line 203).
10. For the perturb-seq assay, the author targeted 55 potential regulatory elements. How were these elements selected? The criteria listed from line 288 to 291 would certainly yield way more than 55 elements. Such a cherry-picked small set of assays would not serve as a valid support of the method and the analysis reported by the current article.

Reviewer #3 (Remarks to the Author): Expert in functional genomics, CRISPRi assays, Perturb-seq, scRNA-seq, and epigenetics; co-reviewed with Reviewer #4

Summary

The authors present MethNet, a bioinformatics pipeline that combines DNA methylation and gene expression data from cancer samples to identify CRE-gene. They highlight the identification of regulatory hubs which contain highly ranked CREs influencing multiple genes and impacting patient survival. They show that the CRE-gene links are enriched in promoter-capture Hi-C loops, demonstrating physical interactions between CREs and their gene targets. The authors validate the functional impact of selected candidate CREs using Perturb-seq. They argue that MethNet provides a valuable resource for understanding complex gene expression mechanisms and gene regulation underlying cancer biology.

Major comments

* From my understanding, the linear modelling approach underlying MethNet essentially uses correlation between methylation state and gene expression levels to identify CRE-gene pairs that covary in methylation state and expression level. This requires variance in both gene expression and methylation state for a CRE-gene connection to be detected. The authors mention in the methods how they filter out genes with low variance in each cancer and fit one model per cancer. This approach might miss very cancer-defining CRE-gene interactions, e.g., if a mutation in a given CRE is always involved in a given cancer there will be very low variance and the model likely won't detect this CRE-gene link. An alternative approach to explore and compare would be to fit a model across all samples and correct for the underlying data structure by applying a similar approach as in Wainberg et al., Nat Genet., 2021 (<https://pubmed.ncbi.nlm.nih.gov/33859415/>), which used generalized least squares (GLS) regression to identify co-essential interactions from CRISPR screens. A similar approach might increase power to detect CRE-gene links that vary between cancers while accounting for non-independence of the samples.

* From the data shown on regulatory hubs it's unclear to me whether the phenomenon of single CREs being linked to multiple genes originates from a CRE regulating multiple genes at the same time vs. a CRE regulating one gene at a time, but different genes in different cancer samples. Since cancer genomes are prone to genomic rearrangements, the 3D landscape could look drastically different between cancer samples leading to rewiring of the CRE-gene regulatory links. Figure 5 together with the observation that hubs are enriched for CTCF sites and have a stronger impact on disease severity are in line with the hypothesis that these CRE control multiple genes in a cell, but this very important piece of the paper could use more supporting data/analyses.

Minor comments

* The sentence from line 35 to 40 is too long and hard to read. Consider splitting into two sentences.

* I don't fully agree with the author's definition of predictive models in the introduction around line 85. I would define a predictive model as a model that was trained to predict a certain outcome from data and if supplied new unseen data, the model can predict the expected outcome.

* Fig 3a: If the formula is shown in the figure, definitions for the terms should also be shown else it's not very informative for the reader.

* A useful analysis for the Perturb-seq screen might be to correlate observed expression changes upon

perturbation of CREs with the predicted regulatory potential/predicted effect size from MethNet.

* In the methods, the authors mention collapsing of proximal CpG probes within 200bp. Even though TCGA lacks chromatin accessibility, it could be useful to intersect the methylation site “clusters” used in MethNet with chromatin accessibility from a cancer cell line for part of ENCODE (e.g; K562, a leukemia derived cell line). Do collapsed CpG probes align with DNase-seq hypersensitive sites (DHS)? Are CpG probes evenly distributed across DHS? How many DHS, which can be viewed as putative regulatory elements, are missed? Similar analyses could be performed with the ENCODE candidate CRE (cCRE) lists, which integrate additional chromatin state information.

* Fulco et al., 2019 (<https://www.nature.com/articles/s41588-019-0538-0>) and Nasser et al., 2021 (<https://www.nature.com/articles/s41586-021-03446-x>) show that 3D interaction frequency decays as a function of distance following roughly $1/\text{distance}$. Using this function to approximate 3D contact in their CRE-gene linking method yielded similar results like using Hi-C data. For computing the MethNet scores, it might be interesting to investigate if the distance effect follows the same behavior.

Reviewer #4 (Remarks to the Author): Expert in functional genomics, CRISPRi assays, Perturb-seq, scRNA-seq, and epigenetics; co-reviewed with Reviewer #3

Reviewer #5 (Remarks to the Author): Expert in cancer epigenomics, epigenetic regulation, Hi-C, methylation, and genetics

The study by Sakellaropoulos and coauthors presents a computational pipeline namely MethNet to discover novel regulatory elements that may not depend the distance to the target genes. Specifically, the authors integrated transcriptome profiling and DNA methylation data from TCGA cancer project, and claimed that a type of hub CREs could be found to regulate multiple genes likely influencing patient survival. The authors eventually applied the promoter-capture Hi-C and scRNA Perturb-seq to test the physical contacts of the hub elements to target genes and whether the elements are truly involved in the control of their target gene expression. Overall, the method appears to be promising while some findings should be further improved.

1) To illustrate that MethNet could capture distal associations, the authors showed two examples on Figure 2e and 2f. It would be nice to test if the 250 kb far CTCF binding site is playing role in the control of IFN γ expression. Whether or not the hypothesis is true, it is worth looking into the correlations between the CTCF site methylation status and the expression of IFN γ in the corresponding TCGA cancer samples. In a similar vein, the hypothesis on the GSTT1 expression control could be examined by CRISPR-mediated DNA (de-)methylation assays. Proper evaluation of the proposed hypotheses may convince the

readers that the MethNet-captured regulatory elements are truly reliable.

2) Related to the results on Figure 4c, whether the authors could further look into the hub CRE regulated genes that also hold potentials to be associated with patient prognosis.

3) In Figure 5, the results on PCDH are problematic. The enhancers in regulating this gene cluster were rather well documented while mostly over developmental stages and in specific tissues such as neurosystem. How did the authors claim that the regulatory hubs discovered from the TCGA cancer samples could be corroborate with these existing findings in given tissues or normal developmental stages. Enhancers in gene regulation are usually cell, tissue-type, or developmental stage specific.

4) In Figure 8a we highlight a regulatory hub...which is unclear where the Figure 8a is located. And vice versa, the Figure 7c/d are missing from the manuscript description.

We would like to thank the reviewers for their thoughtful comments and efforts towards improving our manuscript. We have made changes to address their concerns and incorporate their suggestions and have highlighted the changes within the manuscript.

Please find below our point-by-point rebuttal to the critiques.

Reviewer #1

1. *The analysis between CpG probes and genes has been arbitrarily limited within 1Mbp around the TSSs. Why? What is the molecular basis of such setting?*

In this study, we highlighted long-range gene regulation which we investigated functionally by correlating changes in methylation status of distal elements with changes in gene expression. We hypothesise that the strongest of these associations correspond to direct interactions between CREs and genes and are facilitated via long-range chromatin interactions. Since most regulatory interactions between CREs and genes take place in the context of TADs¹ that are sub-megabase regions of enriched interactions², we opted to perform our analysis in a 1 Mbp region around the promoter. This distance is at least double that of any previously published method that uses methylation to probe the regulatory mechanism. The rationale for using a 1MB radius around the promoter is further supported by our own promoter-capture Hi-C data which shows that although chromatin loops can span super-megabase distances, 94% are in the sub-megabase range with a median value of 311 kbp.

The distribution of loop distances recovered using Arima's pipeline based on ChiCAGO is shown in the following figure. This analysis indicates that looking for methylation/expression associations beyond 1Mbp is most likely going to inflate the false discovery rate of any method, with a very low chance of validating associations using chromatin interaction assays such as promoter capture. The same trend is also seen in our perturb seq experiment (see answer to reviewer 3)

Cumulative distribution of promoter-capture HiC loops distance. Loops were called with Arima pipeline which is based on the CHIGAGO algorithm.

2. ***The authors mentioned that the data of CpG probes within 200bp were collapsed simply by taking averages of the beta values. I don't think this is appropriate. Why 200 but not 100 or 300? Did the authors used a sliding window of 200 bp to scan the whole genome, or did they simply partition the genome into regions of 200bp?***

Clustering neighbouring CpG sites is standard procedure^{3,4} for smoothing technical noise and reducing biological artefacts, such as genetic variants that destroy a CpG sites. Clustering increases the power of the analysis without losing information, since methylation of proximal CpGs is highly correlated⁵ (around 80%). In addition, it is well documented that methylation changes are found in differentially methylated regions (DMR) typically spanning ~100 to 1000 bp regions⁶. The 200 bp window is a standard size for CpG clustering, as it encompasses both typical transcription factor binding sites (CREs) and the length occupied by histones.

MethNet uses methylation as a covariate in a regression model to identify CREs that predict changes in gene expression. Using individual CpG probes rather than clustered probes in our analysis would confound the identification of CREs as being responsible for a regulatory effect. This is because transcription factors do not bind the DNA at single base resolution and can be robust to methylation of a single base. Thus, using the methylation status of a region makes more sense biologically than using individual CpG probes. Moreover, clustering of probes within 200bps provides a good compromise between resolution and robustness.

We opted for an agglomerative clustering approach based exclusively on the distance between probes rather than methylation status in order to avoid heterogeneous clustering across the different cancers. We did not consider a tiling/sliding window approach because the

HumanMethylation450 array that was primarily used to produce the TCGA data only covers 450K CpG sites out of ~20M CpG with heterogeneous probe density.

3. *A simple linear regression model was used to evaluate potential regulation between a CpG cluster and a gene. I don't see novelty of this method design. This model is also too naïve, and the field has long recognized that such simple models can barely recapitulate the complexity of gene regulation programs.*

Most methods developed for identifying interactions between CpG clusters and genes fall under the broad spectrum of what is described as “association mining” which indeed start as simple linear regression models with added filters to control for the false discovery rate. In contrast, we developed a method using a “predictive method”, wherein all possible associations are considered in tandem to account for confounding effects. As such, the false discovery rate was controlled in a principled manner via a regularisation term tuned by cross-validation. Other predictive methods have also been developed but they have a limited scope, typically encompassing a few kilobases around the TSS, in contrast to MethNet which explores a 1Mbp region around the TSS.

We agree with the reviewer that gene expression is not a linear function of the methylation of the surrounding CpG clusters. In fact, we show that linear models can only account for 50% of the variance on average. Our goal is not to model gene regulation but to identify regulatory elements whose methylation status is associated with gene expression changes. To this end, we opted for interpretable models that although under-powered on their own, can be combined with evidence from additional analyses at a later stage to yield associations that are robust to noise. This is important because the number of CpG clusters per gene often exceeds the number of data points available and thus the opportunities of overfitting are high. When combining the evidence, we took into account the performance of the corresponding models as well as the relative position of the CRE to the gene. In particular, when computing the intrinsic regulatory potential of an element we discounted associations from genes in close proximity since these effects are most likely position, rather than element specific.

Below we demonstrate the difference between a naive model, where the regulatory potential of an element is simply the sum of the absolute value of all the associations it participates in, versus our method which predicts interaction hubs (elements with multiple loops) using our promoter-capture Hi-C data.

Chromatin Hub Prediction

4. **Finally, a pan-cancer model was built by taking averages, again, of the coefficients obtained across multiple cancer types. Such an operation simply wiped out the cancer heterogeneity, and this is not a so-called pan-cancer network.**

We agree with the reviewer that the resulting network does not represent the union of all cancer regulatory networks/associations. It is meant instead, to capture the most robust, core regulatory associations that are present across multiple tissues, both under physiological and disease conditions. Cancer samples are a valuable resource for such a task because the resulting heterogeneity can be leveraged as a perturbation of the regulatory mechanism. That is why we do not refer to it as a “pan-cancer” but a “consensus” network. We also highlight this limitation throughout the manuscript, e.g.:

- In discussion of figure 2: *“The context-specificity of the regulatory networks identified are associated with varying degrees of accuracy and reliability across different cancer types”*
- In discussion of figure 7: *“Therefore, although MethNet is useful in identifying high-confidence regulatory hubs, studying their gene regulation network in a tissue-specific manner is also important”*

We changed the following sentence (lines 138-140), where we introduce the concept, to clarify this point:

A consensus network was constructed by aggregating associations across all cancers, so that robust associations found in multiple cancers were ranked higher relative to cancer specific associations, and thus are less likely to correspond to false-positive results

5. **MethNet uses an average of 400,000 (5%) associations per individual cancer to model the expression of every gene.**

We think this point is linked to the following point 6 so we addressed it together.

6. Instead of combining all the data points from all the cancer types, Fig. 2e and 2f should show the positive and negative associations in one cancer type as an example. The associations in different cancer types could be summarized as box plots, for example.

In response to this reviewer’s suggestion, we updated the figure (see below 7e, f). We observe that the signal is robust across all cancers.

7. As shown in Fig. 2a, there are hundreds of CpG clusters around each gene. This would for sure result in high false-positives. Of course, many known cis-regulation would be recovered by MethNet, but this does not address the issue of high false-positive rates.

We agree with the reviewer, and this is a known limitation for every association discovery method. In the case of MethNet, we tried to tackle the problem by aggregating data across multiple cancers in a consensus network. This approach is similar in nature to an empirical-bayes or mixed-effect model where specific instances of an association in a cohort (cancer or tissue) provide evidence for its latent regulatory potential. Associations with strong potential are likely to have strong effects across some or most cohorts and thus would be detected by MethNet, while spurious associations would most likely produce weak or contradicting effects across multiple cohorts.

8. Fig. 3b, what does “promoter of other genes” mean? In the main text, the authors stated this term as “promoters acting as enhancers controlling other distal genes”.

We agree with the reviewer that “Promoter of other genes” might not be a clear description. It refers to “promoters acting as enhancers controlling other distal genes”. In our analysis we focused on distal cis-regulatory elements and therefore did not include the known regulatory effects of methylation of promoters associated with each gene. However, since it has been shown that promoters can act as distal enhancers, we assess the association between methylation at promoters acting as potential distal enhancers. Based on their histone marks and position relative to a TSS, these regions are usually classified as promoters rather than enhancers although they can function as both. To clarify, we changed the label of these regions in figure 3 and 4 to “Promoters acting as Enhancers”.

9. How were the numbers of “potential enrichment” calculated in Fig. 3? Do the negative values mean depletion? The authors reported positive and negative enrichments all together in the main text (line 203).

Yes, the negative values mean depletion. We modified the figures to make this clearer to the reader and clarified this in the legend as well as the method section.

Manuscript update:

In this case [panel c vs b], the basal state represents unbound chromatin and the enrichment score is the average difference in regulatory potential of clusters bound by a specific TF versus being in the unbound condition.

10. For the perturb-seq assay, the author targeted 55 potential regulatory elements. How were these elements selected? The criteria listed from line 288 to 291 would certainly yield way more than 55 elements. Such a cherry-picked small set of assays would not serve as a valid support of the method and the analysis reported by the current article.

Based on the criteria listed from line 288 and 291, 35,000 candidates could have potentially been validated. We could not validate all of these for technical reasons (methylation and expression levels in the cell line we use for testing) and budgetary constraints. Perturb-seq is based on single-cell RNA sequencing, in which the accuracy of the expression levels as well as the detection of differential expression is highly dependent on the number of cells containing a particular sgRNA and the base expression level of the gene. It is known that genes expressed at low level are not easily identified by single-cell RNA sequencing. Therefore, we focused on candidate genes with the highest base expression levels. We also focus on unmethylated genes as we for our CRISPRi we use a dCas9-KRAB-MeCP2 for methylation mediated repression.

The criteria for selecting the candidates to validate are listed in the method sections and we included a table in the supplemental material with the score of every element for every criterion. We also included a new section in the Methods of the paper explaining the criteria in more detail.

We would like to highlight that while we could not validate all the candidate genes in this study, to our knowledge our study is one of the rare studies to endeavour to validate at a genome-wide level methylation/expression association by perturbation experiments, although we agree with the reviewer, the assay is still targeted.

Reviewers #3 & #4

- 1. From my understanding, the linear modelling approach underlying MethNet essentially uses correlation between methylation state and gene expression levels to identify CRE-gene pairs that covary in methylation state and expression level. This requires variance in both gene expression and methylation state for a CRE-gene connection to be detected. The authors mention in the methods how they filter out genes with low variance in each cancer and fit one model per cancer. This approach might miss very cancer-defining CRE-gene interactions, e.g., if a mutation in a given CRE is always involved in a given cancer there will be very low variance and the model likely won't detect this CRE-gene link. An alternative approach to explore and compare would be to fit a model across all samples and correct for the underlying data structure by applying a similar approach as in Wainberg et al., Nat Genet., 2021 (<https://pubmed.ncbi.nlm.nih.gov/33859415/>), which used generalized least squares (GLS) regression to identify co-essential interactions from CRISPR screens. A similar approach might increase power to detect CRE-gene links that vary between cancers while accounting for non-independence of the samples.***

We agree with the reviewer that our approach would conceal the tissue/cancer specific regulatory interactions. Our motivation in this study however, was to uncover interactions that are robust across multiple cancers rather than *cancer-defining CRE-gene interactions*. Thus, we avoided merging data from different TCGA cancers because the TCGA project spanned multiple labs and protocols which inevitably entails the incorporation of batch effects that are hard to regress out. We therefore treated each TCGA-cancer study independently and normalised the coefficients to produce dimensionless units that are comparable across cancers. Consequently,

cancer-defining CRE-gene interactions were detected as low confidence associations which were subsequently ignored because our focus was to identify only robust cross-cancer associations.

There have been several studies that identify tissue-specific methylation^{7,8} and gene⁹ (and GTEx) markers. Our goal was not to recover all cancer specific regulatory associations but a core, robust set of dynamic associations which are recurrent across tissues driven by cancer deregulation. These core CREs are likely to be important for carcinogenic processes in general regardless of the tissue type and could be useful for understanding mechanisms underlying cancer and for identifying targetable hotspots.

With these caveats in mind, we performed an analysis along the lines suggested by the reviewer. In particular, we combined all the samples that have matched gene expression and DNA methylation data. A shortcoming of this analysis is that the effect of missing data propagates across all cancers which results in less associations being tested compared to a per-cancer analysis where missing data in some samples affects only the cancer cohort. Then, for each gene we collected the potential regulatory elements within one megabase, in the same manner that we did for MethNet, and “whitened” the resulting expression-methylation matrix as per Wainberg et al. After that we added extra terms to account for different basal expression levels in different cancer types and for normal vs tumour samples. This makes the results comparable with MethNet, otherwise the GLS method would be prone to identify methylation markers for different tissues and not regulatory elements deregulated by cancer. Finally, we fitted an ordinary least squares regression to predict gene expression and estimated the standard error terms using the covariance pseudo-inverse as suggested by Wainberg et al. To our surprise the results of the 2 methods were not correlated. The spearman correlation of the coefficient for the two methods was -0.03 and for the MethNet score against the GLS-adjusted-p-values was -0.01. Next, we investigated the qualitative predictions of the 2 methods and observed that MethNet can better capture the biology of gene regulation. In particular, when looking at gene promoters, MethNet and GLS predict the canonical silencing of gene expression due to methylation in 80% and 45% of cases, respectively. Moreover, with MethNet, the regulatory effect diminishes strongly as a function of the distance to the TSS, as expected, but GLS effects are much less affected and even seem to tend slightly upwards at great distances. These discrepancies, are due to the fact that the GLS method does not have a regularization term which allows inclusion of many weak associations, even after filtering for low adjusted p-values (0.001).

2. From the data shown on regulatory hubs it's unclear to me whether the phenomenon of single CREs being linked to multiple genes originates from a CRE regulating multiple genes at the same time vs. a CRE regulating one gene at a time, but different genes in different cancer samples. Since cancer genomes are prone to genomic rearrangements, the 3D landscape could look drastically different between cancer samples leading to rewiring of the CRE-gene regulatory links. Figure 5 together with the observation that hubs are enriched for CTCF sites and have a stronger impact on disease severity are in line with the hypothesis that these CRE control multiple genes in a cell, but this very important piece of the paper could use more supporting data/analyses.

MethNet downgrades associations that appear in only a few cancers relative to those that are robustly recovered across multiple cancers. In the figure below, we observe that hubs are enriched in cancer-wide associations relative to non-hub elements. We also computed the average number of associations per cancer for each element and observed that Hubs form on average ~ 3.9 associations per cancer sample, which is significantly more (wilcox p-value $< 2.2 \times 10^{-16}$) than non-hub elements which have on average 1.6 associations per cancer samples.

Moreover, looking at the coefficient of variation (σ/μ), which quantifies how much this number varies across cancers, we observed that of all the CREs with extreme value (top 5%) none were hubs, indicating that their number of associations does not fluctuate a lot.

With regards to the question about the role of CTCF, we think that it is an interesting question but it is outside the scope of this project. We are currently working on a project that investigates the role of CTCF in CRE hub regulation using CTCF mutations.

Minor comments

- 1. The sentence from line 35 to 40 is too long and hard to read. Consider splitting into two sentences.**

We thank the reviewer for pointing this out. We split the sentence to make it clearer.

Changed to:

Nonetheless, noncoding regulatory elements like promoters, enhancers, silencers and structural elements cannot be ignored since variants have been shown to be more likely to contribute to disease than non-synonymous coding variants. Furthermore, noncoding regulatory elements occupy a greater proportion of the genome compared to coding sequences, and they alter the binding capability of factors that are the key drivers of gene regulation.

- 2. I don't fully agree with the author's definition of predictive models in the introduction around in 85. I would define a predictive model as a model that was trained to predict a certain data outcome and if supplied new unseen data, the model can predict the expected outcome.**

We agree with the reviewer that the term “predictive model” is very expansive. It was not our intention to properly define the term but to highlight the contrast between methods that test for specific associations and other methods that primarily model the response variable and recover associations as a by-product.

We changed the term to “regression modelling” to avoid any confusion in the manuscript.

- 3. Fig 3a: If the formula is shown in the figure, definitions for the terms should also be shown else it's not very informative for the reader.**

We removed the formula from the figure. It can be found in the method section with definitions.

- 4. A useful analysis for the Perturb-seq screen might be to correlate observed expression changes upon perturbation of CREs with the predicted regulatory potential/predicted effect size from MethNet.**

We chose to validate our targets with a perturb-seq technology to validate multiple targets in parallel. However, these unparalleled multiplexing capabilities have a number of limitations. In particular, the scRNA assay has low gene coverage relatively to bulk RNA. In our assay, each cell had on average ~4 reads per gene. This sparsity is even more troublesome in our case since we are only interested in the few genes involved in associations identified by MethNet. In addition,

because we are interested in distal regulatory elements, we expect the effect size to be subtle relative to the strong silencing effect of methylation at promoters, and this low-level expression change is likely not identified by scRNA. Because of these limitations, it is hard to perform a holistic analysis and we focused on genes that are adequately expressed (greater than 1 cpm on average, which resulted in 9 validated associations).

To accommodate the reviewer’s request, below we showed the probability of detecting an association identified by MethNet, as a function of distance. In particular, we fit a logistic regression model to predict whether an association is validated out of all possible associations. Here the covariates are (i) the distance between the regulatory element and the TSS of the gene and (ii) whether an association is predicted by MethNet in any cancer or not. We observe that MethNet predicted associations are more likely to be validated by a perturb-seq experiment.

	Estimate	Std. Error	z-value	Pr(> z)
Intercept	-2.88	0.394	-7.31	2.60 10⁻¹³
Distance	-6.02 10⁻⁶	1.35 10⁻⁶	-4.47	7.77 10⁻⁶
MethNet	1.48	0.430	3.45	0.000561

5. In the methods, the authors mention collapsing of proximal CpG probes within 200bp. Even though TCGA lacks chromatin accessibility, it could be useful to intersect

the methylation site “clusters” used in MethNet with chromatin accessibility from a cancer cell line for part of ENCODE (e.g; K562, a leukemia derived cell line). Do collapsed CpG probes align with DNase-seq hypersensitive sites (DHS)? Are CpG probes evenly distributed across DHS? How many DHS, which can be viewed as putative regulatory elements, are missed? Similar analyses could be performed with the ENCODE candidate CRE (cCRE) lists, which integrate additional chromatin state information.

We thank the reviewer for the suggestion. Below we present the results of this analysis. We added the figures to the supplementary material of the paper.

6. *Fulco et al., 2019 (<https://www.nature.com/articles/s41588-019-0538-0>) and Nasser et al., 2021 (<https://www.nature.com/articles/s41586-021-03446-x>) show that 3D interaction frequency decays as a function of distance following roughly $1/\text{distance}$. Using this function to approximate 3D contact in their CRE-gene linking method yielded similar results like using Hi-C data. For computing the MethNet scores, it might be interesting to investigate if the distance effect follows the same behavior.*

The distance of a potential CRE to a gene’s TSS is the most common predictor of regulatory potential. The studies cited by the reviewer show that the regulatory potential scales are inversely proportional to the distance and that most CREs regulate their nearest gene. We investigated this relationship in our study and found that MethNet recovers a similar scaling law as shown in Figure 2d, which uses a rank-based distance. We also investigated the relationship between the regulatory potential and the distance along the chromatin and found that it scales in a non-linear manner as shown in the following figure. Given that at close proximity the regulatory effect of an element can be due to physical proximity rather than its intrinsic potential, when estimating the latter and calling hubs, we regressed out the effect of distance using a non-linear regression model (gam) – see below.

Probability of Detection across Cancers

Reviewer #5

- To illustrate that MethNet could capture distal associations, the authors showed two examples on Figure 2e and 2f. It would be nice to test if the 250 kb far CTCF binding site is playing role in the control of IFN γ expression. Whether or not the hypothesis is true, it is worth looking into the correlations between the CTCF site methylation status and the expression of IFN γ in the corresponding TCGA cancer samples. In a similar vein, the hypothesis on the GSTT1 expression control could be examined by CRISPR-mediated DNA (de-)methylation assays. Proper evaluation of the proposed hypotheses may convince the readers that the MethNet-captured regulatory elements are truly reliable.***

We did not test GSTT1 in our panel of perturb-seq (which validate 55 candidate hubs), since the predicted CRE showed high methylation in our CRISPR cell line and thus could not be assessed. Our CRISPRi experiment is based on MeCP-KRAB inhibition which induces inhibition via both dead Cas9 occupancy and methylation of the CRE, this assay can therefore only evaluate hubs/CRE with low methylation levels in K562 cell lines.

For the case of IFN γ we looked into the data available on the Genome Browser for cell lines that combine expression, methylation and CTCF binding data. In the following figure, we observe that of the 6 cell lines available in 2 of them (GM12878 and H1-hESC) the CRE candidate is methylated

(top arrow) at a 50% level as is the case in our data. In GM12878 CTCF does not bind the CRE (bottom arrow).

At the same time GM12878 is the only cell line where IFN γ is expressed as shown in the following figure. The combination of these observations supports our hypothesis that the silencing potential of this CRE is mediated through a CTCF-regulated mechanism.

IFN γ Region

2. Related to the results on Figure 4c, whether the authors could further look into the hub CRE regulated genes that also hold potentials to be associated with patient prognosis.

We thank the reviewer for the suggestion. To address this, we collected a set of prognostic biomarkers identified by changes in their gene expression as reported by Smith and Sheltzer (2022)¹⁰. In this study, the performed 3 types of analyses

- *Univariate*: where they fit a survival model using a single biomarker
- *Multivariate*: where they accounted for patient age, sex, tumour stage and grade
- *TP53*: where they also included information about the mutational status of TP53

We used the absolute value of the combined Z score (with Stouffer's method) for each gene marker and distributed its effect to all the CRE's associated with it in MethNet, proportionally to their association strength. As a result, we got a prognostic score for each CRE as the mean of the Z-scores of all the genes it is associated with. Below we plot this prognostic score for CRE and Hub elements. We observed that Hubs have significantly higher scores than non-hubs further supporting our hypothesis that they play a crucial role in disease biology. The results are consistent regardless of which model we used to identify the biomarkers.

Meta-Analysis

- 3. In Figure 5, the results on PCDH are problematic. The enhancers in regulating this gene cluster were rather well documented while mostly over developmental stages and in specific tissues such as neurosystem. How did the authors claim that the regulatory hubs discovered from the TCGA cancer samples could be corroborate with these existing findings in given tissues or normal developmental stages. Enhancers in gene regulation are usually cell, tissue-type, or developmental stage specific.**

We agree with the reviewer that the PCDH cluster has been mostly involved in neurodevelopment and neural system related disease and that previous studies supporting the role of the PCDH enhancers in regulating *PCDH* genes were mostly performed in brain or neuronal cell line¹¹⁻¹³. However, *Guo et al.* and *Zhou et al.* data in HEC-1B, a human endometrial adenocarcinoma cell line, showing enhancer-promoter loops and eRNA expression at the HS5-1 enhancer for PCDHA cluster suggest that the PCDH enhancers can also be active and involved with promoter-enhancers chromatin interactions in cancer.

In addition, multiple studies support a role for the PCDH cluster, in acting as both a tumour suppressor or oncogene in brain cancers such as astrocytoma and glioblastoma as well as other solid cancers including colorectal, gastric or prostate cancer¹⁴⁻¹⁶. This is not so surprising as many developmental genes are known to be dysregulated in cancer^{17,18}. Indeed, it has been shown that developmental enhancers can be reprogrammed during carcinogenesis^{19,20}.

Together, these data suggest that the new PCDH regulatory hubs discovered using TCGA cancer data could play a role in both cancer and normal neurodevelopment. However, this hypothesis needs to be confirmed and is the focus of a follow-up project that we are doing in collaboration with Daniele Canzio, an expert in PCDH biology

- 4. In Figure 8a we highlight a regulatory hub...which is unclear where the Figure 8a is located. And vice versa, the Figure 7c/d are missing from the manuscript description.**

We thank the reviewer for pointing out these typos. We have fixed them in the updated manuscript.

References

1. Uyehara, C. M. & Apostolou, E. 3D enhancer-promoter interactions and multi-connected hubs: Organizational principles and functional roles. *Cell Rep.* 112068 (2023) doi:10.1016/j.celrep.2023.112068.
2. Rowley, M. J. & Corces, V. G. Organizational principles of 3D genome architecture. *Nat. Rev. Genet.* **19**, 789–800 (2018).
3. Jaffe, A. E. *et al.* Bump hunting to identify differentially methylated regions in epigenetic epidemiology studies. *Int. J. Epidemiol.* **41**, 200–209 (2012).
4. Condon, D. E. *et al.* Defiant: (DMRs: easy, fast, identification and ANnoTation) identifies differentially Methylated regions from iron-deficient rat hippocampus. *BMC Bioinformatics* **19**, 31 (2018).
5. Eckhardt, F. *et al.* DNA methylation profiling of human chromosomes 6, 20 and 22. *Nat. Genet.* **38**, 1378–1385 (2006).
6. Bock, C. Analysing and interpreting DNA methylation data. *Nat. Rev. Genet.* **13**, 705–719 (2012).
7. Loyfer, N. *et al.* A DNA methylation atlas of normal human cell types. *Nature* **613**, 355–364 (2023).
8. Chakravarthy, A. *et al.* Pan-cancer deconvolution of tumour composition using DNA methylation. *Nat. Commun.* **9**, 3220 (2018).
9. Chen, B., Khodadoust, M. S., Liu, C. L., Newman, A. M. & Alizadeh, A. A. Profiling tumor infiltrating immune cells with CIBERSORT. *Methods Mol. Biol. Clifton NJ* **1711**, 243–259 (2018).

10. Smith, J. C. & Sheltzer, J. M. Genome-wide identification and analysis of prognostic features in human cancers. *Cell Rep.* **38**, 110569 (2022).
11. Guo, Y. *et al.* CTCF/cohesin-mediated DNA looping is required for protocadherin α promoter choice. *Proc. Natl. Acad. Sci. U. S. A.* **109**, 21081–21086 (2012).
12. Jia, Z. *et al.* Tandem CTCF sites function as insulators to balance spatial chromatin contacts and topological enhancer-promoter selection. *Genome Biol.* **21**, 75 (2020).
13. Zhou, Y., Xu, S., Zhang, M. & Wu, Q. Systematic functional characterization of antisense eRNA of protocadherin α composite enhancer. *Genes Dev.* **35**, 1383–1394 (2021).
14. Vega-Benedetti, A. F. *et al.* Clustered protocadherins methylation alterations in cancer. *Clin. Epigenetics* **11**, 100 (2019).
15. El Hajj, N., Dittrich, M. & Haaf, T. Epigenetic dysregulation of protocadherins in human disease. *Semin. Cell Dev. Biol.* **69**, 172–182 (2017).
16. Pancho, A., Aerts, T., Mitsogiannis, M. D. & Seuntjens, E. Protocadherins at the Crossroad of Signaling Pathways. *Front. Mol. Neurosci.* **13**, 117 (2020).
17. Nussinov, R., Tsai, C.-J. & Jang, H. How can same-gene mutations promote both cancer and developmental disorders? *Sci. Adv.* **8**, eabm2059 (2022).
18. An, N., Yang, X., Cheng, S., Wang, G. & Zhang, K. Developmental genes significantly afflicted by aberrant promoter methylation and somatic mutation predict overall survival of late-stage colorectal cancer. *Sci. Rep.* **5**, 18616 (2015).
19. Hnisz, D. *et al.* Convergence of developmental and oncogenic signaling pathways at transcriptional super-enhancers. *Mol. Cell* **58**, 362–370 (2015).

20. Abatti, L. E. *et al.* Epigenetic reprogramming of a distal developmental enhancer cluster drives SOX2 overexpression in breast and lung adenocarcinoma. *Nucleic Acids Res.* **51**, 10109–10131 (2023).

REVIEWER COMMENTS

Reviewer #1 (Remarks to the Author):

My previous comment 2 was not fully addressed. It is still not convincing why the authors arbitrarily cluster the CpG sites within 200-bp regions and simply take averages of the CpG methylation levels. How many CpG sites were located in the 200-bp regions? Are their methylation levels highly correlated? the authors claimed that “methylation of proximal CpGs is highly correlated⁵ (around 80%).” This is not convincing. They need to answer my questions, explicitly, with their data. The whole analyses would not be well-founded if these questions were not well addressed.

For my previous comment 7, to somehow evaluate the potential false positives and also to provide a general picture of the CRE-gene regulatory landscapes, the authors need to provide some basis statistics, for example, how many CRE-gene regulatory logics were inferred for each cancer, how many CREs were inferred for each gene?

For my previous comment 10, I am not convinced at all. Perturb-seq assay with sgRNA pools and single-cell RNA-seq is designed for high-throughput perturbations. Such assays were usually used to test for at least thousands of perturbation assays in parallel. Therefore, I do not understand why only 55 CREs were tested in this study. A reliable validation is critical for this study. Validations of a significantly larger pool of randomly selected CREs is a must to justify the biological relevance and validity of the methodology design.

Reviewer #3 (Remarks to the Author):

Thank you for addressing the raised points. I think the limitations of the approach to detect some tissue/cancer specific regulatory interactions as discussed in major comment 1 should also be discussed in the paper. See the point-to-point response below for more detail. I would also like to emphasize that the code for running MethNet should be publicly available for instance through github.com upon publication (see comments on code).

Point-by-point response

Major comments

1. Thank you for taking your time and trying the suggested GLS approach, although I'm quite surprised by the finding that GLS performs so much poorer. The limitations of their approach that the authors mentioned in their response to my point seem important and should be discussed in the manuscript.

2. These additional analyses showing that hubs are enriched in cancer-wide associations and have more associations per cancer sample compared to non-hub CREs are a valuable addition and could be added to the supplementary figures. The plot in Figure 2b is different from this and doesn't look at hubs specifically, while Figure 4b shows number of associations across cancers if I interpret it correctly.

Minor comments

1-3. Thank you for addressing these points.

4. The motivation for my question was to understand whether high scores predicted by MethNet translate to large effects in your Perturb-seq experiment. In its simplest form that could be a scatter plot showing MethNet scores vs. effect sizes of perturbations on gene expression from the Perturb-seq experiment. In other words, how well does MethNet predict the quantitative contribution of a CRE to a gene's expression.

The presented analysis looks at it from a different angle and doesn't directly investigate quantitative effects on expression. It shows that high MethNet scores lead to higher validation rate, probably due to these having a larger effect on gene expression. I agree with the authors that the sparsity of the data is limiting for these types of analyses, especially if many of the genes of interest have low expression. Since this isn't within the scope of the study, this comment provides food for thought rather than a requirement.

5. Thank you for adding the results from this analysis to the supplementary figures.

6. Thank you for explaining this point

Reviewer #3 (Remarks on code availability):

For a manuscript introducing a new computational method, it's crucial that the code publicly available, for instance through github.com, upon publication. The manuscript in its current state also lacks a code availability statement.

The provided snakemake workflow is a good way to share the code and ensure reproducibility, however it lacks a README file describing software and resource requirements etc. as outlined in Nature's software policy document.

I didn't manage to run workflow because of missing dependencies on my machine. I recommend using a package management system like conda to specify software dependencies, which can also easily be integrated in the snakemake workflow (<https://snakemake.readthedocs.io/en/stable/snakefiles/deployment.html#integrated-package-management>).

Reviewer #4 (Remarks to the Author):

Reviewer #5 (Remarks to the Author):

The revised manuscript by Sakellaropoulos and coworkers has addressed this referee's comments, and some appropriate answers to all the other comments are acceptable. Minor comments remain for this round of review as below.

1. MethNet: a robust approach to identify regulatory hubs and their distal targets from cancer data -> MethNet: a robust approach to identify regulatory hubs and their distal targets in cancer
2. L40-42: In addition, and equally important, epigenetic changes that alter the ability of a transcription factor (TF) to bind a regulatory element can have the same effect. To guide the readers to under this specific point, it would be beneficial to include proper citations. Here's how you can incorporate the suggested citations into the manuscript: e.g. Ref#30, PMID: 28473536; PMID: 33741908; PMID: 38066556.
3. reduced-representation bisulfite sequencing, (RRBS) -> reduced-representation bisulfite sequencing (RRBS).

Response to Reviewers

Reviewer #1

My previous comment 2 was not fully addressed. It is still not convincing why the authors arbitrarily cluster the CpG sites within 200-bp regions and simply take averages of the CpG methylation levels. How many CpG sites were located in the 200-bp regions? Are their methylation levels highly correlated? the authors claimed that “methylation of proximal CpGs is highly correlated (around 80%).” This is not convincing. They need to answer my questions, explicitly, with their data. The whole analyses would not be well-founded if these questions were not well addressed.

We thank the reviewer for their insightful questions. As explained in our previous response, we opted for agglomerative clustering based on genomic distance. The two main parameters of this algorithm are the *cutoff distance* and the *linkage criterion*. A pair of CpG probes are combined into a cluster if their genomic distance is less than or equal to the cutoff distance. In selecting a cutoff distance, we considered 4 options: 100, 200, 500 and 1000 bp. To define the cutoff distance between clusters of probes, we considered the two most common linkage options: *single*, which uses the closest distance, and *complete*, which uses the furthest distance between any pair of probes from the two clusters. Single linkage produces larger clusters because it leads to chaining of smaller clusters as long as a single pair is within the cutoff distance (**Figure 1a**). Such a chaining phenomenon takes place around the 200bp cutoff and it leads to fewer but larger and slightly more heterogenous clusters (**Figure 1b**). That is why in the end we opted for complete linkage, which better controls the number and size of clusters which are also more homogeneous. Increasing the cutoff distance to 500bp did not result in a significant change in the resulting clusters. Moreover, 200bp is the size of regions typically analyzed for transcription factor binding sites in ChIP-seq experiments (eg for motif enrichment) so it aligns well with the resolution of CREs that we were aiming to detect.

Figure 1: a) average size of CpG cluster at different cutoffs (ignoring singletons). b) Number of clusters at different cutoffs and mean coefficient of variations (CV) at different distance cutoff and linkage methods

For my previous comment 7, to somehow evaluate the potential false positives...

As mentioned in our previous response it is hard to accurately estimate the false positive rate because most associations are context dependent and thus it is unclear whether they were missed because (i) they were false positive, or (ii) they were not found in a given tissue or general context. Because of this limitation, MethNet focuses on associations that are present across multiple TCGA studies to construct a consensus network.

This design choice is supported by the fact that associations that are less context specific, like those between genes and CREs in their promoters, are more likely to be recovered across all or multiple cancers. Using this intuition, in **Figure 2** we use the distance between CRE and TSS as a surrogate for true associations (TP = true positive). This is also a common heuristic used to link chromatin regions to their target genes, for example Homer `annotatePeaks.pl` or GREAT. We tested 4 cutoffs from 2kbp to 100kbp. We expect promoter regions to always be involved in gene regulation but the 2kbp cutoff ignores the existence of the enhancers so we expect it to be an underestimate of the true false discovery rate (FDR). Although CREs are expected to be found within a 100kbp of a gene's TSS, most elements at this cutoff will not be involved in gene regulation. Thus, the 100kbp cutoff represents an overestimate of the FDR. Using this surrogate, we compute the FDR as a function of how often it is recovered by MethNet across multiple cancers. The figure shows that the number of cancers supporting an association can be used to mitigate FDR, especially for close proximity associations.

Figure 2: FDR score as a function of how often an association is recovered by MethNet's initial steps.

More direct experimental evidence for the accuracy and false positive rate of MethNet are provided by the promoter-capture HiC analysis in the main text (**Figure 6**).

... and also to provide a general picture of the CRE-gene regulatory landscapes, the authors need to provide some basis statistics, for example, how many CRE-gene regulatory logics were inferred for each cancer, how many CREs were inferred for each gene

We added an extra supplemental figure (**Figure S1**), included here for convenience (**Figure 3**) with the requested basic statistics of CRE and associations identified per cancer study. We included statistics both in terms of absolute numbers and in terms of fraction of the potential CREs/associations that could be identified. On average, the first steps of MethNet recovers 417,018.6 associations per cancer (standard deviation of 106,910.8) which corresponds to 5% of all possible associations within a 1Mbp radius. These association involved 168,389.6 CREs on average (standard deviation of 22,714.58) or 54% of all the possible CREs considered. However, on a per gene basis, only 74 CREs, or 21% of all possible regulators were included with any strength for a given cancer. On the other hand, CREs on average are associated with 2.5 genes per cancer and 10.5 genes across all cancers. The distribution that these numbers summarize are shown in the figure.

Figure 3: Basic statistic of MethNet cancer specific regulatory networks. A) Number of recovered associations and CREs per TCGA study. B) Fraction of the potential and absolute number of CRE (associations) per gene across TCGA studies. C) Fraction of the potential and absolute number of genes (associations) per CRE across TCGA studies

For my previous comment 10, I am not convinced at all. Perturb-seq assay with sgRNA pools and single-cell RNA-seq is designed for high-throughput perturbations. Such assays were usually used to test for at least thousands of perturbation assays in parallel. Therefore, I do not understand why only 55 CREs were tested in this study. A reliable validation is critical for this study. Validations of a significantly larger pool of randomly selected CREs is a must to justify the biological relevance and validity of the methodology design.

In our experiment, we analyzed 55 CREs by sequencing 25,000 cells (500 cells per target). The largest perturb-seq study to date [PMID:32483332] analyzed 2000 targets by sequencing 200,000 cells (100 cells per target). Our approach requires more cells per target because we are looking for weaker effects. We recognize that targeting 55 CREs is not a comprehensive validation of our findings, but the number was selected based on financial constraints. Scaling the pool of sgRNA to 1000 as the reviewer suggests would increase the cost of library design from \$10,000 to \$14,000, per Collecta's estimate. Most importantly however, it would raise the sequencing cost by a factor of 20. Given the current cost of single cell sequencing this would cost around \$300,000.

▶ Mar 23 10:37 AM	Single-cell (10X Genomics), sample	RNAseq	Quantity: 11.0	Unit Price: \$2,756.00	Total: \$2,756.00	Billing Status: Paid	Work Status: Completed	 Single cell RNASeq								
▶ Mar 23 10:38 AM	Hashtag/CITE-Seq Library Preparation Labor + Reagents		Quantity: -2.0	Unit Price: \$172.00	Total: \$344.00	Billing Status: Paid	Work Status: Completed	 Single cell RNASeq								
▶ Mar 27 01:06 PM	S4 200 cycle flow v1.5 Illumina NovaSeq 6000	cell	Quantity: 0.09	Unit Price: \$14,685.00	Total: \$1,321.65	Billing Status: Paid	Work Status: Completed	 
Reviewers #3 & #4:

Thank you for taking your time and trying the suggested GLS approach, although I'm quite surprised by the finding that GLS performs so much poorer. The limitations of their approach that the authors mentioned in their response to my point seem important and should be discussed in the manuscript.

As pointed out in our previous response, the poor performance of GLS was also a surprise to us. We think GLS' poor performance can be explained in terms of the increased complexity introduced by the combination of multiple cancers which is not counter-balanced by an adequate increase in the sample size. That was partially the reason why, in developing MethNet, we opted for smaller but less complex datasets and introduced all the post-fitting steps to reduce confounding results.

As requested, we added the following paragraph to our discussion:

“Our motivation in this study, was to uncover interactions that are robust across multiple cancers rather than cancer-defining CRE-gene interactions. There have been several studies that identify tissue-specific methylation and gene markers. MethNet does not aim to recover all cancer specific regulatory associations, but instead identifies a core, robust set of dynamic associations which are recurrent across tissues and revealed by cancer deregulation. These core CREs are likely to be important for carcinogenic processes in general, regardless of the tissue type and could be useful for understanding mechanisms underlying cancer and for identifying targetable hotspots. We therefore treated each TCGA-cancer study independently, and 5normalized the coefficients to produce dimensionless units that are comparable across cancers. Cancer-defining CRE-gene interactions were detected as low confidence associations which were effectively ignored because our focus was to identify robust cross-cancer CREs.”

These additional analyses showing that hubs are enriched in cancer-wide associations and have more associations per cancer sample compared to non-hub CREs are a valuable addition and could be added to the supplementary figures. The plot in Figure 2b is different from this and doesn't look at hubs specifically, while Figure 4b shows number of associations across cancers if I interpret it correctly

We added the figure used in our response as panels in **Supplemental Figure S4** and adapted the figure legend and the main text accordingly. The figure and the legend are included here for convenience as **Figure 5**.

Figure 3: Enrichment of Hubs in MethNet associations and open chromatin regions. For the open chromatin analysis, we overlapped MethNet CREs with Dnase peaks of K562 cells downloaded from ENCODE. A) Distributions of average associations per cancer for CREs that

For a manuscript introducing a new computational method, it's crucial that the code publicly available, for instance through github.com, upon publication. The manuscript in its current state also lacks a code availability statement.

The provided snakemake workflow is a good way to share the code and ensure reproducibility, however it lacks a README file describing software and resource requirements etc. as outlined in Nature's software policy document.

I didn't manage to run workflow because of missing dependencies on my machine. I recommend using a package management system like conda to specify software dependencies, which can also easily be integrated in the snakemake workflow (<https://snakemake.readthedocs.io/en/stable/snakefiles/deployment.html#integrated-package-management>).

We thank the reviewers for testing our pipeline and providing feedback for improvements. We have updated the code to add conda support and a README file to explain the structure of the code as requested. The code is hosted at GitHub: <https://github.com/TeoSakel/MethNet>. We tested the pipeline in our Linux cluster using Slurm scheduler. In case the reviewer does not have access to high performance computing, they could test it with fewer cancers and/or genes as a proof of principle.

We have also added the main results of our run as supplementary data (**Supplementary Data**) so that people without access to an HPC cluster can identify and analyze associations and/or CREs within regions of interest to them.

Reviewer #5:

The revised manuscript by Sakellaropoulos and coworkers has addressed this referee's comments, and some appropriate answers to all the other comments are acceptable. Minor comments remain for this round of review as below.

We thank the reviewer for their valuable comments and contributions. We incorporated all suggestions in the final manuscript.

REVIEWER COMMENTS

Reviewer #1 (Remarks to the Author):

I am quite disappointed that the authors did not answer my question directly, again. Let me repeat my question for the third time. How heterogeneous are the CpG methylation levels within the CREs? Again, a simple statement “methylation of proximal CpGs is highly correlated (around 80%)” is not informative enough to address this question. How proximal are they? What is considered “highly correlated”? What is around 80%? The authors need to show that for most of their CREs, the methylation levels are highly homogenous. Otherwise, taking averages of multiple CpG sites within the CREs is simply not right.

I am glad that Fig. S1 is provided, finally. However, as I have suspected, Figure S1B shows that for a number of genes, almost all the candidate CREs (fraction of CREs close to 1) were predicted by MethNet to regulate the gene expression. These CREs are located within quite a large window of 1Mbp radius of the TSS. Again, as I have mentioned previously, this raises a serious issue of potential false positives of CRE-gene associations.

Reviewer #3 (Remarks to the Author):

Trying to run the MethNet software took some time and we had to try several snakemake versions and commands. It would be great if the authors could be more specific on this in the GitHub readme file. Otherwise, authors have sufficiently addressed my concerns and provided acceptable answers to my questions. If the comments of other reviewers are addressed appropriately as well, I recommend accepting the manuscript for publication.

Reviewer #3 (Remarks on code availability):

see above

Reviewer #4 (Remarks to the Author):

Response to Reviewer 1

I am quite disappointed that the authors did not answer my question directly, again. Let me repeat my question for the third time. How heterogeneous are the CpG methylation levels within the CREs? Again, a simple statement “methylation of proximal CpGs is highly correlated (around 80%)” is not informative enough to address this question. How proximal are they? What is considered “highly correlated”? What is around 80%? The authors need to show that for most of their CREs, the methylation levels are highly homogenous. Otherwise, taking averages of multiple CpG sites within the CREs is simply not right.

We apologize for the confusion, let us restate our response in simple terms.

In our first response we pointed out that clustering neighboring CpG sites is very common practice as it increases the detection power with negligible cost in resolution. **In our second response**, we pointed out that several metrics can be used to evaluate the performance of clustering besides homogeneity and we explored different options before settling on the 200bp cutoff. The following paragraphs expand on these points and focus on the points re-raised by the reviewer here.

Clustering neighboring CpG sites is a very common practice in the field because methylation is homogeneous within regulatory elements and varies smoothly along neighboring CpGs. This is evident both from the fact that a typical methylation analysis workflow, like the one suggested by Bioconductor (the most popular bioinformatics repository), aims to identify differentially methylated *regions* (DMRs) and the fact that most regulatory elements affected by methylation are defined as discrete genomic regions, e.g. CpG islands and promoters, and not over single CpG sites. Given this biological evidence, any apparent heterogeneity *within* CREs is usually treated as noise, either technical or biological, which is effectively suppressed by aggregating the constituent CpGs.

Figure 1 and 2 demonstrate these facts. **Figure 1** is from the original publication of Bumphunter¹, a very popular method for identifying DMRs. It demonstrates the smooth variation of methylation along neighboring CpG sites. Notice that the correlation length is in the order of 1000bp, much larger than the 200bp used in MethNet. **Figure 2** is from our own data and shows the homogeneity of DNA methylation of CRE regions at single molecule level.

[Figure 1 was taken from the published article by Jaffe AE, Murakami P, Lee H, Leek JT, Fallin MD, Feinberg AP, Irizarry RA. Bump hunting to identify differentially methylated regions in epigenetic epidemiology studies. *Int J Epidemiol.* 2012 Feb;41(1):200-9. doi: 10.1093/ije/dyr238 which has been **Redacted]**

[Figure 2 Redacted]

Several metrics can be used to evaluate the performance of clustering besides homogeneity. In our response to the 2nd round of revisions, we demonstrated the trade-offs for 3 of those metrics:

- cluster size (resolution)
- number of clusters
- coefficient of variation (homogeneity)

as a function of the 2 main parameters of agglomerative clustering: linkage method and cutoff distance. Here, in the interest of brevity and clarity, we will focus only on homogeneity, since this is the main focus of the reviewer's question. We only analyze the parameters that were used for MethNet, namely complete linkage and cutoff at 200 bp.

It is important to note that Illumina's HumanMethylation450k array takes advantage of the aforementioned homogeneity of regulatory regions, by selecting a few representative CpG sites, in order to maximize genome coverage. Because of the sparsity of CpG probes, most of the CRE candidates (72%) are "singletons", i.e. they consist of a single probe, and thus the homogeneity concern is only relevant for 28% of the CRE candidates.

For these 28% of candidates, as already stated in the second round of review, we quantified homogeneity using the coefficient of variation (CV) which is defined as the ratio of standard deviation over the mean of a distribution (σ/μ). In our case, the distribution is the ensemble of CpG probes defining a cluster within a single patient sample.

Figure 3A shows the distribution of CV of all CpG clusters with more than 2 CpG sites averaged across all samples. Because interpreting the CV of bound distributions can be challenging, we also include the shape of Beta distributions (the natural choice for values bound between 0 and 1) for different CV and mean values. The CV and mean values for the distribution are taken from the bins shown.

We also include the distribution of the standard deviations of the same clusters in **Figure 3B** which is another common metric that quantifies cluster heterogeneity and is easier to interpret. We observe that most clusters, have a very low standard deviation of less than 0.1.

Figure 3: Distribution of CpG Cluster Heterogeneity for clusters with more than 2 CpG probes. **A)** Distribution of Cluster Sizes. We used clusters with more than 2 CpG probes in order to be able to calculate mean and standard deviation. **B)** Distribution of cluster CV. We grouped the clusters into 3 groups, as indicated by the dashed lines (cutoff 0.2 and 0.6). The ratio of each group as well as the average (μ) and standard deviation (σ) are shown. To illustrate the effect of CV in the cluster distribution, we include a sketch of a Beta distribution with the same mean value and average CV for each cluster (the standard deviations are not necessarily the same because CV is not a linear operator). **C)** Distribution of cluster standard deviation. **D)** Joint distribution of average standard deviation and fractional methylation of the clusters across all samples. The points are colored by their average CV. High CV in red corresponds to the high group in panel B (2.1% of the clusters).

I am glad that Fig. S1 is provided, finally. However, as I have suspected, Figure S1B shows that for a number of genes, almost all the candidate CREs (fraction of CREs close to 1) were predicted by MethNet to regulate the gene expression. These CREs are located within quite a large window of 1Mbp radius of the TSS. Again, as I have mentioned previously, this raises a serious issue of potential false positives of CRE-gene associations.

It was not our intention to withhold Figure S1, we just did not realize that it was requested in their initial comments. **We completely agree with the reviewer that including almost all association, regardless of distance, would dramatically inflate the false positive rate but this is not what we did.** Figure S1 shows all associations which are enriched in cancer-specific associations (as shown in Figure 2b of the main manuscript) that as we acknowledged in the main manuscript are prone to be false positive. For example, in the Discussion section we state “Cancer-defining CRE-gene

interactions were detected as low confidence associations which were effectively ignored because our focus was to identify robust cross-cancer CREs". **The distribution of high-confidence associations does not show the same behavior.** Below we expand on our answer.

Figure S1B shows a wide range of value (the x-axis is on log scale) that is centered closer to the 10% than the 100% mark. The reviewer's comment is probably motivated by the bimodality of the distributions, with a secondary mode appearing around 80% for some cancers. As shown in **Figure 4A and 4B** of the current document, this bimodality is driven by genes with relatively few CRE candidates in their neighborhood so it represents a discrete "artifact". Specifically, most genes have a ratio of 10% of MethNet CREs to all possible potential candidates, while 10-20% of the genes have a high ratio (greater than 75%) driven mostly by genes with few candidates CREs (these are concentrated near the beginning of the x-axis).

This artifact originally motivated us to define a MethNet score that controlled for the different number of candidates per gene (the $\log \frac{1}{k_g}$ part of the formula in the Methods section of the manuscript). In the manuscript and throughout the revision process, we have taken great care to highlight and address issues of false positives, which is a major challenge of any GWAS-like study. However, the reviewer's comment focuses on the initial steps of analysis, while ignoring the strength of the associations and the aggregation across multiple cancers that come from later steps in the analysis. By focusing on associations that MethNet assigns a positive score to define high-confidence CREs (meaning that they contribute more than expected), the problematic mode disappears as shown in **Figures 4C and 4D** of the current document. We would like to re-emphasize that our analysis does not focus on individual genes or cancers but aggregates evidence across all the genes and cancers in order to identify robust elements.

Figure 4: Joined distribution of the number of CRE candidates per gene and the fraction of CREs selected by MethNet. The left panels are the distributions of all associations recovered by MethNet, while the right panels are the distributions of the associations with positive MethNet score (which defines “Methnet” CREs). The bottom panels summarize the top ones by quartile as shown by the common legend in the middle. MethNet associations where most (>75%) of the candidates were identified as CREs correspond to Q2 and Q1 which are not present (or negligible) among high-confidence CREs.

We agree with the reviewer that a 1Mbp radius is a wide range and most candidates within that range would be irrelevant. In fact, in our 2nd response to the reviewer, we used a 1Mb range as a proxy for false discovery. However, we and other labs have shown that long range interactions do play a role in gene regulation²⁻⁴ so searching for them is not misguided if the false discovery rate can be controlled. In fact, searching for distal intergenic CREs adds to the novelty of our study since no other method looks for these, as we discuss in the introduction of the main text.

Finally, in order to fully address the relationship between selection probability and distance we include **Figure 5A** that shows the probability of forming an association as a function of distance. MethNet associations are strongly affected by distance as demonstrated by the sharp drop in the probability of selection. Positive associations are more specific (less associations called) across all distances and more sensitive to the effect of distance (sharper drop in the probability of selecting associations at greater distance, which are probably false positive).

The fact that MethNet strongly enriches for more proximal associations is not trivial, as shown in a comparison with another method (**Figure 5B**). This was highlighted in response to reviewers 3 and 4 during the first round of revisions. It is interesting to note that GLS ranks distal association higher which shows that generic methodologies that do not account for the biological context of the application are sensitive to high false discovery rates. Thus, we conclude that MethNet mitigates the effect of distance better than generic methodologies.

Figure 5: Effect of distance in probability of detection. **A)** Probability of selecting associations by MethNet. "Initial" refers to all association from MethNet's initial step, "Final" refers to associations that contribute to MethNet CRE scores. **B)** Comparison of MethNet with GLS method suggested by Reviewers 3 & 4. MethNet ranks higher associations between gene and promoters which is the canonical function of methylation.

Response to Reviewer 3 and 4

Trying to run the MethNet software took some time and we had to try several snakemake versions and commands. It would be great if the authors could be more specific on this in the GitHub readme file. Otherwise, authors have sufficiently addressed my concerns and provided acceptable answers to my questions. If the comments of other reviewers are addressed appropriately as well, I recommend accepting the manuscript for publication.

We thank the reviewers for their comments and feedback that enhanced the quality of the manuscript and pipeline. As mentioned in the readme file of the GitHub page, we used **snakemake 8.5.3**. Since GitHub provides an issue tracker, we will happily address their or any future specific issues raised through this more appropriate platform.

References

1. Jaffe, A. E. *et al.* Bump hunting to identify differentially methylated regions in epigenetic epidemiology studies. *International Journal of Epidemiology* **41**, 200–209 (2012).
2. Roldán, E. *et al.* Locus ‘decontraction’ and centromeric recruitment contribute to allelic exclusion of the immunoglobulin heavy-chain gene. *Nat Immunol* **6**, 31–41 (2005).
3. Snetkova, V. & Skok, J. A. Enhancer talk. *Epigenomics* **10**, 483–498 (2018).
4. Uyehara, C. M. & Apostolou, E. 3D enhancer-promoter interactions and multi-connected hubs: Organizational principles and functional roles. *Cell Rep* 112068 (2023) doi:10.1016/j.celrep.2023.112068.

REVIEWERS' COMMENTS

Reviewer #1 (Remarks to the Author):

I have no further questions. I suggest to include some of the responses to my comments in the Discussion section.

Response to Reviewer #1

I have no further questions. I suggest to include some of the responses to my comments in the Discussion section.

We thank the reviewer for their constructive comments throughout the review process. We added the following paragraphs (and accompanying Supplementary Figures) to the Discussion section as suggested:

Any method that identifies regulatory elements by linking gene expression with methylation, must deal with the problem of spurious correlations. This issue is exacerbated by the fact that we consider long range (up to 1Mbp) associations and that nearby methylation sites tend to be correlated and can act synergistically. To address the confounding effect of correlated methylation sites, we clustered probes within 200bp into a single variable. Clustering neighboring CpG sites is standard procedure for smoothing technical noise and reducing biological artefacts, such as genetic variants that destroy a CpG sites. Clustering increases the power of the analysis without losing information, since methylation of proximal CpGs is highly correlated. In addition, it is well documented that methylation changes are found in differentially methylated regions (DMR) typically spanning ~100 to 1000 bp regions. The 200 bp window is a standard size for CpG clustering, as it encompasses both typical transcription factor binding sites and the length occupied by histones. We evaluated the performance of the clustering using three metrics: mean cluster size, number of clusters and coefficient of variation (Figure S10).

Although 1Mbp range interactions are important for gene regulation, testing all possible promoter-CRE pairs is prone to produce high false discovery rates. We addressed this issue by combining data across genes and cancers in a statistically principled manner. In particular, we used elastic-net regression, tuned with cross-validation, to promote sparsity within every cancer and then pooled the resulting associations across cancers based on their predictive strength while accounting for known confounding factors (see Methods: MethNet score). This multi-level approach significantly reduced the number of identified CREs per gene compared to a naïve analysis of variance with lasso-penalty (Figure S11).